# Descent Steps of a Relation-Aware Energy Produce Heterogeneous Graph Neural Networks

**Hongjoon Ahn**[1]*, **Yongyi Yang**[2]*, **Quan Gan** [3], **Taesup Moon** [1]† **and David Wipf** [3]†

[1] ECE/IPAI/ASRI/INMC, Seoul National University, [2] University of Michigan, [3] Amazon Web Services

`{hong0805, tsmoon}@snu.ac.kr, yongyi@umich.edu,`
`quagan@amazon.com, davidwipf@gmail.com`

## Abstract

Heterogeneous graph neural networks (GNNs) achieve strong performance on node classification tasks in a semi-supervised learning setting. However, as in the simpler homogeneous GNN case, message-passing-based heterogeneous GNNs may struggle to balance between resisting the oversmoothing that may occur in deep models, and capturing long-range dependencies of graph structured data. Moreover, the complexity of this trade-off is compounded in the heterogeneous graph case due to the disparate heterophily relationships between nodes of different types. To address these issues, we propose a novel heterogeneous GNN architecture in which layers are derived from optimization steps that descend a novel relation-aware energy function. The corresponding minimizer is fully differentiable with respect to the energy function parameters, such that bilevel optimization can be applied to effectively learn a functional form whose minimum provides optimal node representations for subsequent classification tasks. In particular, this methodology allows us to model diverse heterophily relationships between different node types while avoiding oversmoothing effects. Experimental results on 8 heterogeneous graph benchmarks demonstrate that our proposed method can achieve competitive node classification accuracy. The source code of our algorithm is available at https://github.com/hongjoon0805/HALO.

## 1 Introduction

Graph structured data, which contains information related to entities as well as the interactions between them, arise in various applications such as social networks or movie recommendations. In a real world scenario, these entities and interactions are often multi-typed, such as *actor*, *director*, *keyword*, and *movie* in a movie review graph [20] or *authors*, *papers*, *terms*, and *venue* in an academic network [31]. Hence the relationships between entities can potentially be much more complex than within a traditional homogeneous graph scenario, in which only a single node and edge type exists. To this end, *heterogeneous* graphs have been proposed as a practical and effective tool to systematically deal with such multi-typed graph-structured data. With the advent of graph neural networks (GNN) for instantiating deep models that are sensitive to complex relationships between data instances, it is natural that numerous variants have been developed to explicitly handle heterogeneous graphs [2, 5, 7, 10, 17, 18, 29, 33, 37], with promising performance on downstream tasks such as node classification or link prediction.

In this vein, message-passing GNNs represent one of the most widely-adopted heterogeneous architectures, whereby a sequence of type-specific graph propagation operators allow neighboring nodes to share and aggregate heterogeneous information [7, 29, 33]. However, in some cases where

---

*Work completed during an internship at the AWS Shanghai AI Lab.

†Co-corresponding author.

36th Conference on Neural Information Processing Systems (NeurIPS 2022).

long-range dependencies exist between nodes across the graph, stacking too many layers to reflect a larger neighborhood may contribute to a well-known *oversmoothing* problem, in which node features converge to similar, non-discriminative embeddings [16, 22]. In the case of homogeneous graphs, this oversmoothing problem has been addressed (among other ways) using modified GNN architectures with layers patterned after the unfolded descent steps of some graph-regularized energy [8, 36]. Provided minimizers of this energy remain descriminative for the task at hand, arbitrarily deep networks can be trained without risk of oversmoothing, with skip connections between layers organically emerging to favor node-specific embeddings. However, extending this approach to handle heterogeneous graphs is not straightforward, largely because node/edge heterogeneity (*e.g.*, node-specific numbers of classes or feature dimensions, multiple types of relationships between nodes) is not easily reflected in the vanilla energy functions adopted thus far for the homogeneous case. In particular, existing energy functions are mostly predicated on the assumption of graph homophily, whereby neighboring nodes have similar labels. But, in the heterogeneous case, this assumption is no longer realistic because the labels between nodes of different types may exhibit complex relationships.

To this end, we first introduce a novel heterogeneous GNN architecture with layers produced by the minimization of a relation-aware energy function. This energy produces regularized heterogeneous node embeddings, relying on relation-dependent compatibility matrices as have been previously adopted in the context of belief propagation on heterogeneous graphs, e.g., ZooBP [6]. Secondly, we derive explicit propagation rules for our model layers by computing gradient descent steps along the proposed energy, with quantifiable convergence conditions related to the learning rate being considered as well. Furthermore, we provide interpretations of each component of the resulting propagation rule, which consists of skip-connections, linearly transformed feature aggregation, and a self-loop transformation. With respect to experiments, our contributions are twofold. First, we show that our algorithm outperforms state-of-the-art baselines on various benchmark datasets. And secondly, we analyze the the effectiveness of using the compatibility matrix and the behavior of the unfolding step as the number of propagations increases.

## 2 Existing Homogeneous GNNs from Unfolded Optimization

Consider a *homogeneous* graph $\mathcal{G} = \{\mathcal{V}, \mathcal{E}\}$, with $n = |\mathcal{V}|$ nodes and $m = |\mathcal{E}|$ edges. We define $\mathbf{L} \in \mathbb{R}^{n \times n}$ as the Laplacian of $\mathcal{G}$, meaning $\mathbf{L} = \mathbf{D} - \mathbf{A}$, in which $\mathbf{D}$ and $\mathbf{A}$ are degree and adjacency matrices, respectively. Unfolded GNNs incorporate graph structure by attaining optimized embeddings $\mathbf{Y}^* \in \mathbb{R}^{n \times d}$ that are functions of some adjustable weights $\mathbf{W}$, i.e., $\mathbf{Y}^* \equiv \mathbf{Y}^*(\mathbf{W})$,[3] where $\partial \mathbf{Y}^*(\mathbf{W})/\partial \mathbf{W}$ is computable by design. These embeddings are then inserted within an application-specific loss

$$\ell_{\boldsymbol{\theta}}(\boldsymbol{\theta}, \mathbf{W}) \triangleq \sum_{i=1}^{n'} \mathcal{D}\big(g\left[\boldsymbol{y}_i^*(\mathbf{W}); \boldsymbol{\theta}\right], \boldsymbol{t}_i\big), \tag{1}$$

in which $g : \mathbb{R}^d \to \mathbb{R}^c$ is some differentiable node-wise function with parameters $\boldsymbol{\theta}$ and $c$-dimensional output tasked with predicting ground-truth node-wise targets $\boldsymbol{t}_i \in \mathbb{R}^c$. Additionally, $\boldsymbol{y}_i^*(\mathbf{W})$ is the $i$-th row of $\mathbf{Y}^*(\mathbf{W})$, $n' < n$ is the number of labeled nodes (we assume w.l.o.g. that the first $n'$ nodes are labeled), and $\mathcal{D}$ is a discriminator function, *e.g.*, cross-entropy for node classification, as will be our focus. Given that $\mathbf{Y}^*(\mathbf{W})$ is differentiable by construction, we can optimize $\ell_{\boldsymbol{\theta}}(\boldsymbol{\theta}, \mathbf{W})$ via gradient descent (over *both* $\boldsymbol{\theta}$ and $\mathbf{W}$ if desired) to obtain our final predictive model.

At a conceptual level, what differentiates GNNs inspired by unfolded optimization is how $\mathbf{Y}^*(\mathbf{W})$ is obtained. Specifically, these node embeddings are chosen to be the minimum of a lower-level, graph-regularized energy function [3, 19, 23, 36, 38, 40] (a similar idea has also recently been applied to complete graphs within Tranformer architectures [35]). Orignally inspired by [39], the most common selection for this energy is given by

$$\ell_{\mathbf{Y}}(\mathbf{Y}) \triangleq \|\mathbf{Y} - f(\mathbf{X}; \mathbf{W})\|_{\mathcal{F}}^2 + \lambda \mathrm{tr}\left[\mathbf{Y}^\top \mathbf{L} \mathbf{Y}\right], \tag{2}$$

where $\lambda > 0$ is a trade-off parameter, $\mathbf{Y} \in \mathbb{R}^{n \times d}$ is a matrix of trainable embeddings, meaning $d$-dimensional features across $n$ nodes, and $f(\mathbf{X}; \mathbf{W})$ is a base model parameterized by $\mathbf{W}$ that produces initial target embeddings using the $d_0$-dimensional node features $\mathbf{X} \in \mathbb{R}^{n \times d_0}$. Minimization of (2) over $\mathbf{Y}$ results in new node embeddings $\mathbf{Y}^*$ characterized by a balance between similarity

---

[3]For brevity, we will frequently omit this dependency on $\mathbf{W}$ when the context is clear.

with $f(\mathbf{X}; \mathbf{W})$ and smoothness across graph edges as favored by the trace term $\text{tr}\left[\mathbf{Y}^\top \mathbf{L}\mathbf{Y}\right] = \sum_{\{i,j\}\in\mathcal{E}} \|\mathbf{y}_i - \mathbf{y}_j\|_2^2$, where $\mathbf{y}_i$ is $i$-th row of $\mathbf{Y}$. While (2) can be solved in a closed-form as

$$\mathbf{Y}^*(\mathbf{W}) = \arg\min_{\mathbf{Y}} \ell_{\mathbf{Y}}(\mathbf{Y}) = (\mathbf{I} + \lambda\mathbf{L})^{-1} f(\mathbf{X}; \mathbf{W}), \tag{3}$$

for large graphs this analytical solution is not practically computable. Instead, we may form initial node embeddings $\mathbf{Y}^{(0)} = f(\mathbf{X}; \mathbf{W})$, and then apply gradient descent to converge towards the minimum, with the $k$-th iteration given by

$$\mathbf{Y}^{(k)} = \mathbf{Y}^{(k-1)} - \alpha\left[(\lambda\mathbf{L} + \mathbf{I})\mathbf{Y}^{(k-1)} - f(\mathbf{X}; \mathbf{W})\right], \tag{4}$$

in which $\frac{\alpha}{2}$ is the step size. Given that $\mathbf{L}$ is generally sparse, computation of (4) can leverage efficient sparse matrix multiplications. Combined with the fact that the loss is strongly convex, and provided that $\alpha$ is chosen suitably small, we are then guaranteed to converge towards the unique global optima with efficient gradient steps, meaning that for some $K$ sufficiently large $\mathbf{Y}^{(K)}(\mathbf{W}) \approx \mathbf{Y}^*(\mathbf{W})$.

Critically, $\mathbf{Y}^{(K)}(\mathbf{W})$ remains differentiable with respect to $\mathbf{W}$, such that we may substitute this value into (1) in place of $\mathbf{Y}^*(\mathbf{W})$ to obtain an entire pipeline that is end-to-end differentiable w.r.t. $\mathbf{W}$. Moreover, per the analyses from [19, 23, 36, 38, 40], the $k$-th *unfolded* iteration of (4) can be interpreted as an efficient form of GNN layer. In fact, for certain choices of the step size, and with the incorporation of gradient pre-conditioning and other reparameterization factors, these iterations can be exactly reduced to popular canonical GNN layers such as those used by GCN [14], APPNP [8], and others. However, under broader settings, we obtain unique GNN layers that are naturally immune to oversmoothing, while nonetheless facilitating long-range signal propagation across the graph by virtue of the anchoring effect of the underlying energy function. In other words, regardless of how large $K$ is, these iterations/layers will converge to node embeddings that adhere to the design criteria of (2). This is the distinct appeal of GNN layers motivated by unfolded optimization, at least thus far in the case of homogeneous graphs.

## 3 New Heterogeneous GNN Layers via Unfolding

In this section, we explore novel extensions of unfolded optimization to handle heterogeneous graphs, which, as mentioned previously, are commonly encountered in numerous practical application domains. After introducing our adopted notation, we first describe two relatively simple, intuitive attempts to accommodate heterogeneous node and edge types. We then point out the shortcomings of these models, which are subsequently addressed by our main proposal: a general-purpose energy function and attendant proximal gradient steps that both (i) descend the aforementioned energy, and (ii) in doing so facilitate flexible node- and edge-type dependent message passing with non-linear activations where needed. The result is a complete, interpretable heterogeneous GNN (HGNN) architecture with layers in one-to-one correspondence with the minimization of a heterogeneous graph-regularized energy function.

Before we begin, we introduce some additional notation. An undirected heterogeneous graph $\mathcal{G} = (\mathcal{V}, \mathcal{E})$ is a collection of node types $\mathcal{S}$ and edge types $\mathcal{T}$ such that $\mathcal{V} = \bigcup_{s\in\mathcal{S}} \mathcal{V}_s$ and $\mathcal{E} = \bigcup_{t\in\mathcal{T}} \mathcal{E}_t$. Here $\mathcal{V}_s$ denotes a set of $n_s = |\mathcal{V}_s|$ nodes of type $s$, while $\mathcal{E}_t$ represents a set of edge type $t$. Furthermore, we use $\mathcal{T}_{ss'}$ to refer to the set of edge types connecting node types $s$ and $s'$. Note that $\mathcal{T}$ includes both the canonical direction, $t \in \mathcal{T}_{ss'}$, and the inverse direction, $t_{\text{inv}} \in \mathcal{T}_{s's}$, that corresponds with a type $t$ edge.[4]

### 3.1 First Attempts at Heterogeneous Unfolding

A natural starting point for extending existing homogeneous models to the heterogeneous case would be to adopt the modified energy

$$\ell_{\mathbf{Y}}(\mathbf{Y}) \triangleq \sum_{s\in\mathcal{S}} \left[\frac{1}{2}\|\mathbf{Y}_s - f(\mathbf{X}_s; \mathbf{W}_s)\|_{\mathcal{F}}^2 + \frac{\lambda}{2}\sum_{s'\in\mathcal{S}}\sum_{t\in\mathcal{T}_{ss'}} \text{tr}\left(\mathbf{Y}^\top \mathbf{L}_t \mathbf{Y}\right)\right], \tag{5}$$

in which $\mathbf{L}_t$ is the graph Laplacian involving all edges of relation type $t$ and $\mathbf{W}_s$ parameterizes a type-specific transformation of node representations. Here, we have simply introduced type-specific

---

[4]For example, if $t \in \mathcal{T}_{ss'}$ is ("paper"-"author"), then $t_{\text{inv}} \in \mathcal{T}_{s's}$ is ("author"-"paper")

transformations of the initial node representations, as well as adding a separate graph regularization term for each relation type $t$. However, this formulation is actually quite limited as it can be reduced to the equivalent energy

$$\ell_{\mathbf{Y}}(\mathbf{Y}) \equiv \frac{1}{2}\left\|\mathbf{Y} - \widetilde{f}(\widetilde{\mathbf{X}}; \mathcal{W})\right\|_{\mathcal{F}}^2 + \frac{\lambda}{2}\mathrm{tr}\left(\mathbf{Y}^\top \mathbf{C}\mathbf{Y}\right), \quad \text{with } \mathbf{C} = \sum_{s\in\mathcal{S}}\sum_{s'\in\mathcal{S}}\sum_{t\in\mathcal{T}_{ss'}} \mathbf{L}_t, \quad (6)$$

in which $\widetilde{\mathbf{X}}$ represents the original node features concatenated with the node type, and $\widetilde{f}$ is some function of the augmented features parameterized by $\mathcal{W} \triangleq \{\mathbf{W}_s\}_{s\in\mathcal{S}}$. Note that by construction, $\mathbf{C}$ will necessarily be positive semi-definite, and hence we may conclude through (6) that (5) defaults to the form of a standard quadratically-regularized loss, analogous to (2), that has often been applied to graph signal processing problems [12]. Therefore the gradient descent iterations of this objective will closely mirror the existing homogeneous unfolded GNN architectures described in Section 2, and fail to capture the nuances of heterogeneous data.

To break the symmetry that collapses all relation-specific regularization factors into $\mathrm{tr}\left(\mathbf{Y}^\top \mathbf{C}\mathbf{Y}\right)$, a natural option is to introduce $t$-dependent weights as in

$$\ell_{\mathbf{Y}}(\mathbf{Y}) \triangleq \sum_{s\in\mathcal{S}}\left[\frac{1}{2}\|\mathbf{Y}_s - f(\mathbf{X}_s; \mathbf{W}_s)\|_{\mathcal{F}}^2 + \frac{\lambda}{2}\sum_{s'\in\mathcal{S}}\sum_{t\in\mathcal{T}_{ss'}}\mathrm{tr}\left(\mathbf{Y}^\top \mathbf{L}_t \mathbf{Y}\mathbf{M}_t\right)\right], \quad (7)$$

in which the set $\{\mathbf{M}_t\}$ is trainable. While certainly more expressive than (5), this revision is nonetheless saddled with several key limitations: (i) Unless additional constraints are included on $\{\mathbf{M}_t\}$ (e.g., PSD, etc.), (7) may be unbounded from below; (ii) While $\mathbf{M}_t$ may be asymmetric, w.l.o.g. the penalty can be equivalently expressed with symmetric weights, and hence the true degrees of freedom are limited;[5] (iii) This model could be prone to overfitting, since during training it could be that $\mathbf{M}_t \to 0$ in which case the graph-based regularization is turned off;[6] and (iv) Because $\mathrm{tr}\left[\mathbf{Y}^\top \mathbf{L}_t \mathbf{Y}\mathbf{M}_t\right] = \sum_{(i,j)\in\mathcal{E}_t}(\boldsymbol{y}_{si} - \boldsymbol{y}_{s'j})^\top \mathbf{M}_t(\boldsymbol{y}_{si} - \boldsymbol{y}_{s'j})$, the energy function (7) is symmetric w.r.t. the order of the nodes in the regularization term, and therefore any derived message passing must also be symmetric, *i.e.*, we cannot exploit asymmetric penalization aligned with heterogeneous relationships in the data.

Given then that both (5) and (7) have notable shortcomings, it behooves us to consider revised criteria for selecting a suitable heterogeneous energy function. We explore such issues next.

## 3.2 A More Expressive Alternative Energy Function

As an entry point for developing a more flexible class of energy functions that is sensitive to nuanced relationships between different node types, we consider previous work developing various flavors of both label and belief propagation [6, 27, 34, 39]. In the more straightforward setting of homogeneous graphs under homophily conditions, label propagation serves as a simple iterative process for smoothing known training labels across edges to unlabeled nodes in such a way that, for a given node $i$, the predicted node label $\hat{\boldsymbol{t}}_i$ will approximately match the labels of neighboring nodes $\mathcal{N}_i$, *i.e.*, $\hat{\boldsymbol{t}}_i \approx \hat{\boldsymbol{t}}_j$ when $j \in \mathcal{N}_i$. An analogous relationship holds when labels are replaced with beliefs [27].

However, in broader regimes with varying degrees of heterophily, it is no longer reasonable for such a simple relationship to hold, as neighboring nodes may be more inclined to have *dissimilar* labels. To address this mismatch, a compatibility matrix $\mathbf{H}$ must be introduced such that now label propagation serves to instantiate $\hat{\boldsymbol{t}}_i \approx \hat{\boldsymbol{t}}_j \mathbf{H}$ for $j \in \mathcal{N}_i$ [34] (here we treat each $\hat{\boldsymbol{t}}_j$ as a row vector). If each predicted label $\hat{\boldsymbol{t}}_i$ is approximately a one-hot vector associated with class membership probabilities, then the $(k,l)$-th element of $\mathbf{H}$ roughly determines how a node of class $k$ influences neighbors of class $l$. And if we further extend to heterogeneous graphs as has been done with various forms of belief propagation [6], it is natural to maintain a unique (possibly non-square) compatibility matrix

---

[5]Note that $\mathbf{M}_t = \frac{1}{2}[(\mathbf{M}_t + \mathbf{M}_t^\top) + (\mathbf{M}_t - \mathbf{M}_t^\top)]$ for any weight matrix $\mathbf{M}_t$. Since $\boldsymbol{y}^\top(\mathbf{M}_t - \mathbf{M}_t^\top)\boldsymbol{y} = 0$ for any $\boldsymbol{y}$ and $\mathbf{M}_t$, the penalty actually only depends on the symmetric part $(\mathbf{M}_t + \mathbf{M}_t^\top)$.

[6]For example, suppose the node features of training nodes are highly correlated with the labels, or actually are the labels (as in some prior label propagation work). Then the model could just learn $\mathbf{M}_t = 0$ and achieve perfect reconstruction on the training nodes to produce zero energy, and yet overfit since graph propagation is effectively turned off.

$\mathbf{H}_t$ for each relation type $t$, with the goal of finding predicted labels (or beliefs) of each node type satisfying $\hat{\boldsymbol{t}}_{si} \approx \hat{\boldsymbol{t}}_{s'j}\mathbf{H}_t$ for $s, s' \in \mathcal{S}$, $t \in \mathcal{T}_{ss'}$, and $(i, j) \in \mathcal{E}_t$. From a practical standpoint, each such $\mathbf{H}_t$ can be estimated from the data using the statistics of the labels (or beliefs) of nodes sharing edges of type $t$.

Returning to our original goal, we can use similar intuitions to guide the design of an appropriate regularization factor for learning heterogenous graph node representations (as required by HGNNs). Specifically, given $s, s' \in \mathcal{S}$, $t \in \mathcal{T}_{ss'}$, and $(i, j) \in \mathcal{E}_t$, we seek to enforce $\boldsymbol{y}_{si}\mathbf{H}_t \approx \boldsymbol{y}_{s'j}$, which then naturally motivates the energy

$$\ell_{\mathbf{Y}}(\mathbf{Y}) \triangleq \sum_{s \in \mathcal{S}} \left[ \frac{1}{2}||\mathbf{Y}_s - f(\mathbf{X}_s; \mathbf{W}_s)||_{\mathcal{F}}^2 + \frac{\lambda}{2} \sum_{s' \in \mathcal{S}} \sum_{t \in \mathcal{T}_{ss'}} \sum_{(i,j) \in \mathcal{E}_t} ||\boldsymbol{y}_{si}\mathbf{H}_t - \boldsymbol{y}_{s'j}||_2^2 \right], \quad (8)$$

where $\lambda$ denotes a trade-off parameter, $\mathbf{X}_s \in \mathbb{R}^{n_s \times d_{0s}}$ represents $d_{0s}$-dimensional initial features and $\mathbf{Y}_s \in \mathbb{R}^{n_s \times d_s}$ is the embedding of $d_s$-dimensional features on $n_s$ nodes for node type $s$. Note that $\boldsymbol{y}_{si}$ and $\boldsymbol{y}_{s'j}$ are the $i$-th and $j$-th row of $\mathbf{Y}_s$ and $\mathbf{Y}_{s'}$, respectively. Additionally, $\mathbf{H}_t \in \mathbb{R}^{d_s \times d_{s'}}$ denotes a compatibility matrix that matches the dimension of two different node embeddings (*i.e.*, $\boldsymbol{y}_{si}$ and $\boldsymbol{y}_{s'j}$) via linear transformation for edge type $t$. Incidentally, the $\mathbf{H}_t$-dependent term in (8) resembles a form of score function often used to learn knowledge graph embeddings [13].

The energy function (8) exhibits the following two characteristics:

- The first term prefers that the embedding $\mathbf{Y}_s$ should be close to that from $f(\mathbf{X}_s; \mathbf{W}_s)$.
- The second term prefers that for two nodes of type $s$ and $s'$ connection by an edge of type $t$, the embeddings $\boldsymbol{y}_{si}\mathbf{H}_t$ and $\boldsymbol{y}_{s'j}$ should be relatively close to one another.

Even when the dimensions of two embeddings are the same (*i.e.*, $d_s = d_{s'}$), $\mathbf{H}_t$ still serves an important role in allowing the embeddings to lie within different subspaces to obtain compatibility.

### 3.3 Analysis and Derivation of Corresponding Descent Steps

The objective (8) can be written as a matrix form as follows:

$$\ell_{\mathbf{Y}}(\mathbf{Y}) = \sum_{s \in \mathcal{S}} \left[ \frac{1}{2}||\mathbf{Y}_s - f(\mathbf{X}_s; \mathbf{W}_s)||_{\mathcal{F}}^2 \right.$$
$$\left. + \frac{\lambda}{2} \sum_{s' \in \mathcal{S}} \sum_{t \in \mathcal{T}_{ss'}} \mathrm{tr}((\mathbf{Y}_s\mathbf{H}_t)^\top \mathbf{D}_{st}(\mathbf{Y}_s\mathbf{H}_t) - 2(\mathbf{Y}_s\mathbf{H}_t)^\top \mathbf{A}_t\mathbf{Y}_{s'} + \mathbf{Y}_{s'}^\top \mathbf{D}_{s't_{\mathrm{inv}}}\mathbf{Y}_{s'}) \right], \quad (9)$$

where $\mathbf{A}_t$ denotes the adjacency matrix for an edge type $t$, and $\mathbf{D}_{st}$ denotes the type-$t$ degree matrix of type-$s$ nodes.

The closed-form optimal point of (9) is provided by the following result; see the Supplementary Materials for the proof.

**Lemma 3.1.** *The unique solution $\mathbf{Y}^*(\mathcal{W}, \mathcal{H})$ minimizing (9) satisfies*

$$vec(\mathbf{Y}^*(\mathcal{W}, \mathcal{H})) = (\mathbf{I} + \lambda(\mathbf{Q} - \mathbf{P} + \mathbf{D}))^{-1}vec(\widetilde{f}(\widetilde{\mathbf{X}}; \mathcal{W})), \quad (10)$$

*where $vec(\mathbf{B}) = [\mathbf{b}_1^\top, ..., \mathbf{b}_n^\top]^\top$ for matrix $\mathbf{B}$, $\mathcal{H} \triangleq \{\mathbf{H}_t\}_{t \in \mathcal{T}}$ denotes the set of all compatibility matrices, and we have defined the matrices $\mathbf{P}$, $\mathbf{Q}$, and $\mathbf{D}$ as*

$$\mathbf{P} = \begin{bmatrix} \mathbf{P}_{11} & ... & \mathbf{P}_{1|\mathcal{S}|} \\ ... & ... & ... \\ \mathbf{P}_{|\mathcal{S}|1} & ... & \mathbf{P}_{|\mathcal{S}||\mathcal{S}|} \end{bmatrix}; \mathbf{P}_{ss'} = \sum_{t \in \mathcal{T}_{ss'}} ((\mathbf{H}_t^\top + \mathbf{H}_{t_{inv}}) \otimes \mathbf{A}_t)$$

$$\mathbf{Q} = \bigoplus_{s \in \mathcal{S}} \mathbf{Q}_s; \mathbf{Q}_s = \sum_{s' \in \mathcal{S}} \sum_{t \in \mathcal{T}_{ss'}} (\mathbf{H}_t\mathbf{H}_t^\top \otimes \mathbf{D}_{st}), \mathbf{D} = \bigoplus_{s \in \mathcal{S}} \mathbf{I} \otimes \mathbf{D}_s.$$

*Here $\otimes$ denotes the Kronecker product, $\bigoplus_{i=1}^n \mathbf{A}_i = diag(\mathbf{A}_1, ..., \mathbf{A}_n)$ denotes a direct sum of $n$ square matrices $\mathbf{A}_1, ..., \mathbf{A}_n$, and $\mathbf{D}_s \triangleq \sum_{s' \in \mathcal{S}} \sum_{t \in \mathcal{T}_{ss'}} \mathbf{D}_{s't}$ represent a sum of degree matrices over all node types $s' \in \mathcal{S}$ and all edge types $t \in \mathcal{T}_{ss'}$.*

We can interpret this solution as transforming $\widetilde{f}(\widetilde{\mathbf{X}}; \mathcal{W})$ into an embedding that has both local and global information of the graph structure [39]. Therefore, this can be treated as an appropriate graph-aware embedding for carrying out a downstream task such as node classification.

However, for practically-sized graph data, computing the inverse $(\mathbf{I} + \lambda(\mathbf{Q} - \mathbf{P} + \mathbf{D}))^{-1}$ in (10) is prohibitively expensive. To resolve this issue, similar to the homogeneous case we instead apply gradient descent to (9) over $\mathbf{Y}_s$ for all node types $s \in \mathcal{S}$ to approximate $\mathbf{Y}^*(\mathcal{W}, \mathcal{H})$. Starting from the initial point $\mathbf{Y}_s^{(0)} = f(\mathbf{X}_s; \mathbf{W}_s)$ for each $s \in \mathcal{S}$, several steps of gradient descent are performed to obtain an appropriate embedding for carrying out a downstream task. Since all the intermediate steps of gradient descent are differentiable features w.r.t $\mathcal{W}$ and $\mathcal{H}$, we can just plug-in the result of performing $K$ iterations of gradient descent, $\mathbf{Y}_s^{(K)}$, into the (18).

In order to get $\mathbf{Y}_s^{(K)}$ using gradient descent, we first compute the gradient of (9) w.r.t $\mathbf{Y}_s$:[7]

$$\nabla_{\mathbf{Y}_s} \ell_{\mathbf{Y}}(\mathbf{Y}) = (\mathbf{I} + \lambda \mathbf{D}_s)\mathbf{Y}_s - f(\mathbf{X}_s; \mathbf{W}_s) + \lambda \sum_{s' \in \mathcal{S}} \sum_{t \in \mathcal{T}_{ss'}} \left( \mathbf{D}_{st} \mathbf{Y}_s (\mathbf{H}_t \mathbf{H}_t^\top) - \mathbf{A}_t \mathbf{Y}_{s'} (\mathbf{H}_t^\top + \mathbf{H}_{t_{\mathrm{inv}}}) \right). \tag{11}$$

Note that if the condition number of (11) is large, the convergence speed of gradient descent can be slow [21]. To reduce the convergence time effectively, we use the Jacobi preconditioning technique [1], rescaling the gradient step using $\widetilde{\mathbf{D}}_s^{-1} \triangleq (\mathbf{I} + \lambda \mathbf{D}_s)^{-1}$. Then iteration $k + 1$ of gradient descent on $\mathbf{Y}_s$ can be computed as

$$\mathbf{Y}_s^{(k+1)} =$$
$$\underbrace{(1 - \alpha)\mathbf{Y}_s^{(k)}}_{(a)} + \alpha \widetilde{\mathbf{D}}_s^{-1} \Big[ \underbrace{f(\mathbf{X}_s; \mathbf{W}_s)}_{(b)} + \lambda \sum_{s' \in \mathcal{S}} \sum_{t \in \mathcal{T}_{ss'}} \Big( \underbrace{\mathbf{A}_t \mathbf{Y}_{s'}^{(k)}(\mathbf{H}_t^\top + \mathbf{H}_{t_{\mathrm{inv}}})}_{(c)} - \underbrace{\mathbf{D}_{st} \mathbf{Y}_s^{(k)}(\mathbf{H}_t \mathbf{H}_t^\top)}_{(d)} \Big) \Big], \tag{12}$$

where $\alpha$ denotes a step size. The above expression can be considered as a forward propagation rule for the $k$-th layer in a GNN model, and when re-expressed on a per-node basis shares several commonalities with the node-wise updates of R-GCN [29], a commonly-used heterogeneous GNN variant that is motivated in a completely different fashion. In (12), the terms (a), (b), (c), and (d) can be interpreted as follows:

**Terms (a) & (b)** Each term can be treated as a skip connection from the previous layer $\mathbf{Y}_s^{(k)}$ and the input features $f(\mathbf{X}_s; \mathbf{W}_s)$, respectively. Unlike some previous work [15, 18] that adopted skip-connections as a heuristic solution to resolve the oversmoothing problem [16, 22], similar to unfolded GNNs in the homogeneous case [8, 36], the skip connections in our expression are naturally derived from the gradient descent step for minimizing the energy function.

**Term (c)** This term accumulates the propagated feature vectors of neighboring nodes. The matrix $\mathbf{H}_t^\top + \mathbf{H}_{t_{\mathrm{inv}}}$ serves as edge-type specific transformations in a bi-directional way. Note that all matrices in $\mathcal{H}$ are shared across all propagation layers, and as a result, the number of parameters does not increase as the propagation proceeds.

**Term (d)** This term is a self-loop transformation that depends on the edge type $t$. The matrix $\mathbf{H}_t \mathbf{H}_t^\top$ introduces an edge-type specific transformation in a uni-directional way, and the matrix $\mathbf{D}_{st}$ reflects the strength of the self-loop transformation relying on edge type $t$. Note that, unlike prior work [18, 29], we use edge-type specific degree and transformation matrices, and these are also shared across all GNN layers.

We reiterate that all components of (12) have naturally emerged from the gradient descent procedure for minimizing the underlying energy function. And in terms of convergence, we have the following result; again, the proof is deferred to the Supplementary Materials:

**Theorem 3.2.** *The iterations (12) are guaranteed to monotonically converge to the unique global minimum of (8) provided that*

$$\alpha < \frac{2 + 2\lambda d_{min}}{1 + \lambda(d_{min} + \sigma_{max})}, \tag{13}$$

---

[7]This form of the gradient implicitly assumes that for every directed edge there is an inverse edge pointing in the opposite direction. However, the more general case can also be handled with appropriate modification of the gradient expression.

*where $d_{min}$ is the minimum diagonal element of $\mathbf{D}$ and $\sigma_{max}$ is a maximum eigenvalue of matrix $(\mathbf{Q} - \mathbf{P})$.*

**Time Complexity**  Although seemingly complicated, the time complexity of executing (12) once is $O(md + nd^2)$, where $d = \max\{d_s | s \in \mathcal{S}\}$. In this way the complexity is on the same level as R-GCN [29], and it can be efficiently implemented through off-the-shelf sparse matrix libraries.

### 3.4  Generalization to Nonlinear Activations

In various message-passing GNNs [9, 14, 18, 29], node-wise ReLU activations are applied to the intermediate features during the forward propagation step. In our setting, by adding additional regularization or constraints to our original energy function and using proximal gradient descent methods [24], such nonlinear activations can also be naturally derived without any heuristic modifications on our propagation step. Similar ideas have been applied to simpler homogeneous GNN models [36]. More specifically, let $\phi_s : \mathbb{R}^{d_s} \to \mathbb{R}$ be an arbitrary convex function of node embeddings. Then our optimization objective can be modified the minimization of

$$\ell_{\mathbf{Y}}(\mathbf{Y}) + \sum_{s \in \mathcal{S}} \sum_{i=1}^{n_s} \phi_s(\boldsymbol{y}_{si}). \tag{14}$$

Instead of directly minimizing (14) via vanilla gradient descent, we employ the proximal gradient descent (PGD) method [24] as follows. We first denote the relevant proximal operator as

$$\mathbf{prox}_{\phi}(\boldsymbol{v}) \triangleq \arg \min_{\boldsymbol{y}} \frac{1}{2}||\boldsymbol{v} - \boldsymbol{y}||_2^2 + \phi(\boldsymbol{y}). \tag{15}$$

Then PGD iteratively minimizes (14) by computing

$$\bar{\mathbf{Y}}_s^{(k+1)} := \mathbf{Y}_s^{(k)} - \alpha \widetilde{\mathbf{D}}_s^{-1} \nabla_{\mathbf{Y}_s^{(k)}} \ell_{\mathbf{Y}}(\mathbf{Y}) \tag{16}$$

$$\mathbf{Y}_s^{(k+1)} := \mathbf{prox}_{\phi_s}(\bar{\mathbf{Y}}_s^{(k+1)}). \tag{17}$$

For our purposes, we set $\phi_s(\boldsymbol{y}) \triangleq \sum_{i=1}^{d_s} \mathcal{I}_\infty[y_i < 0]$, in which $\mathcal{I}_\infty$ is an indicator function that assigns an infinite penalty to any $y_i < 0$. In this way, the corresponding proximal operator becomes $\mathbf{prox}_{\phi_s}(\boldsymbol{v}) = \mathrm{ReLU}(\boldsymbol{v}) = \max(\mathbf{0}, \boldsymbol{v})$, in which the maximum is applied to each dimension independently.

---

**Algorithm 1** HALO algorithm

---

**Require:** $\mathcal{G}$: Heterogeneous graph dataset
**Require:** $\lambda, \alpha$: Hyperparamters, $K$ : Number of unfolding steps, $E$: Number of epochs
  Randomly initialize $\mathbf{H}_t \in \mathcal{H}$, $\mathbf{W}_s \in \mathcal{W}$, and $\boldsymbol{\theta}_s \in \Theta$
  **for** $e = 1, \cdots, E$ **do**
    Set $\mathbf{Y}_s^{(0)} = f(\mathbf{X}_s; \mathbf{W}_s)$ for all $s \in \mathcal{S}$
    **for** $k = 0, \cdots, K - 1$ **do**
      Compute (16) using (12), and (17) to get $\mathbf{Y}_s^{(k+1)}$ for each $s \in \mathcal{S}$
    **end for**
    Compute $\ell_\Theta(\Theta, \mathcal{W}, \mathcal{H})$ using $\mathbf{Y}_s^{(K)}$
    Optimize all $\mathbf{W}_s \in \mathcal{W}$, $\mathbf{H}_t \in \mathcal{H}$, and $\boldsymbol{\theta}_s \in \Theta$ using SGD.
  **end for**

---

### 3.5  The Overall Algorithm

After performing $K$ iterations of (12) with the proximal operator in (17), we obtain $\mathbf{Y}^{(K)}(\mathcal{W}, \mathcal{H})$, and analogous to (1), the meta-loss function for downstream tasks becomes

$$\ell_\Theta(\Theta, \mathcal{W}, \mathcal{H}) = \sum_{s \in \mathcal{S}'} \sum_{i=1}^{n_s'} \mathcal{D}\big(g_s[\boldsymbol{y}_{si}^{(K)}(\mathcal{W}, \mathcal{H}); \boldsymbol{\theta}_s], \boldsymbol{t}_{si}\big), \tag{18}$$

where $\mathcal{S}'$ denotes a set of labeled node types, $g_s : \mathbb{R}^{d_s} \to \mathbb{R}^{c_s}$ denotes a node-wise function parameterized by $\boldsymbol{\theta}_s$, $\Theta \triangleq \{\boldsymbol{\theta}_s\}_{s \in \mathcal{S}'}$ denotes a set of parameters of $g_s$, and $\boldsymbol{t}_{si} \in \mathbb{R}^{c_s}$ is a ground-truth node-wise target.[8] Finally, we optimize (18) over $\mathcal{W}$, $\mathcal{H}$, and $\Theta$ via typical iterative optimization method such as stochastic gradient descent (SGD).

---

[8]Note that w.l.o.g., we assume that only the first $n_s' < n_s$ type-$s$ nodes have labels for training.

Table 1: Results on HGB (left) and knowledge graphs (right). The results are averaged over 5 runs.

| Dataset | DBLP | IMDB | ACM | Freebase | Avg. | | Dataset | AIFB | MUTAG | BGS | AM | Avg. |
|---------|------|------|-----|----------|------|---|---------|------|-------|-----|-----|------|
| Metric | Accuracy (%) | | | | | | Metric | Accuracy (%) | | | | |
| R-GCN[29] | 92.07 | 62.05 | 91.41 | 58.33 | 75.97 | | Feat[26] | 55.55 | 77.94 | 72.41 | 66.66 | 68.14 |
| HAN[33] | 92.05 | 64.63 | 90.79 | 54.77 | 75.56 | | WL[30, 4] | 80.55 | 80.88 | 86.20 | 87.37 | 83.75 |
| HGT[11] | 93.49 | 67.20 | 91.00 | 60.51 | 78.05 | | RDF2Vec[28] | 88.88 | 67.20 | 87.24 | 88.33 | 82.91 |
| Simple-HGN[18] | 94.46 | 67.36 | 93.35 | **66.29** | 80.37 | | R-GCN[29] | 95.83 | 73.33 | 83.10 | 89.29 | 85.39 |
| HALO | **96.30** | **76.20** | **94.33** | 62.06 | **82.22** | | HALO | **96.11** | **86.17** | **93.10** | **90.20** | **91.40** |

We dub the model obtained by the above procedure as HALO for ***Heterogeneous Architecture Leveraging Optimization***, and Algorithm 1 summarizes the overall process.

## 4 Experiments

We evaluate HALO on node or entity classification tasks from the heterogeneous graph benchmark (HGB) [18] as well as several knowledge graph datasets [29], and compare with state-of-the-art baselines. Furthermore, we later evaluate w.r.t. ZooBP under settings that permit direct comparisons, and we carry out additional empirical analyses and ablations.

### 4.1 Node Classification on Benchmark Datasets

The dataset descriptions are as follows:

**HGB** contains 4 node classification datasets, which is our focus herein. These include: DBLP, IMDB, ACM, and Freebase. DBLP and ACM are citation networks, IMDB is a movie information graph, and Freebase is a large knowledge graph. Except for Freebase, all datasets have node features (for Freebase, we set the node types as the node features). For more details, please refer to [18].

The **knowledge graph** benchmarking proposed in [29] is composed of 4 datasets: AIFB, MUTAG, BGS, AM. The task is to classify the entities of target nodes. AIFB and MUTAG are relatively small-scale knowledge graphs, while BGS and AM are larger scale, *e.g.*, the AM dataset has more than 1M entities. For more details, please refer to [29].

In every experiment, we chose $f(\mathbf{X}_s; \mathbf{W}_s)$ and $g_s(\boldsymbol{y}_s, \boldsymbol{\theta}_s)$ as linear functions for all $s \in \mathcal{S}$, and when the dimension $d_{s0}$ is different across node types, $f(\mathbf{X}_s; \mathbf{W}_s)$ is naturally constructed to align the dimensions of each feature. All models and experiments were implemented using PyTorch [25] and the Deep Graph Library (DGL) [32]. For the details on the hyperparameters and other experimental settings, please see the Supplementary Materials.

Table 1 shows the results on node classification tasks, where the best performance is highlighted in bold. The left side of the table shows the results on HGB, including baselines R-GCN [29], HAN [33], HGT [11], and Simple-HGN [18]. The right side shows the analogous results on the knowledge graph datasets, where we also include comparisons with R-GCN [29], hand-designed feature extractors (Feat) [26], Weisfeiler-Lehman kernels (WL), and RDF2Vec embeddings [28]. The Avg. column in each table represents the average of the results across the four datasets. In the table, HALO achieves the highest average accuracy, and outperforms the baselines on 7 of 8 datasets.

### 4.2 Comparison with ZooBP

We next compare HALO with ZooBP [6], which provided motivation for our use of compatibility matrices and represents a natural baseline to evaluate against when possible. ZooBP is a belief propagation method for heterogeneous graphs, and therefore, the number of classes per node type should be pre-defined for every node. Consequently, running ZooBP on HGB or the knowledge graph datasets in which only a single node type has labels (and therefore defined classes), *e.g.*, DBLP, is not directly feasible. To that end, to compare with ZooBP, we modified the DBLP and Academic [31] datasets so that all node types have labels. Namely, in both graphs, we removed the "Term" and "Venue" node types and only utilized the "Paper" and "Author" node types. Moreover, since the original datasets only have the class labels for the "Author" type, we set the class labels for the "Paper" type as the corresponding "Venue" of the paper. Thus, the 20 and 18 Venues (specified in the Supplementary Materials) become the class labels for the "Paper" nodes in DBLP and Academic, respectively. Furthermore, we also considered a simpler setting in which the number of class labels for the "Paper" nodes (denoted as $k$) is four, by using the 4 categories (*i.e.*, ML, DB, DM, IR) of the Venues as the labels for the nodes. We note the number of class labels for the "Author" nodes

Table 2: Comparison with ZooBP

| Dataset | DBLP-reduced | | Academic-reduced | |
|---|---|---|---|---|
| Metric | Accuracy (%) | | | |
| Node types | Author / Paper / All | | Author / Paper / All | |
| # Classes of "Paper" | $k = 4$ | $k = 20$ | $k = 4$ | $k = 18$ |
| ZooBP[6] | 63.00 / 63.60 / 63..47 | 62.00 / 19.53 / 28.47 | 79.63 / 73.43 / 75.97 | 81.76 / 16.08 / 42.94 |
| Ours | **76.50 / 64.13 / 66.79** | **78.50 / 33.33 / 43.00** | **96.63 / 93.89 / 94.93** | **97.81 / 84.73 / 90.18** |

was always set to 4 (corresponding to the Venue categories), and the resulting datasets are denoted as DBLP-reduced and Academic-reduced, respectively.[9]

For choosing the *fixed* compatibility matrix required by ZooBP (as a label propagation-like method, ZooBP has no mechanism for learning), when $k = 4$, we set $\mathbf{H}_t = \mathbf{I}_{k \times k} - \frac{1}{4} \mathbf{1}_{k \times k}$, where $\mathbf{1}_k$ is a $k \times k$ square matrix in which all elements are 1. When $k = 18$ or $k = 20$, we set the matrix considering the correlation between the "Author" and "Paper" labels. For example, for the "AAAI" Paper class, we set the matrix element value corresponding to the "ML" Author class higher than other Author classes, since "AAAI" venue and "ML" category are highly correlated. For further details about reducing the number of classes for the "Paper" nodes, setting the compatibility matrices, and the exact compatibility matrices that we used, please refer to the Supplementary Materials. Moreover, for setting the output layer $g_s(\boldsymbol{y}_s, \boldsymbol{\theta}_s)$ in our case, we set $g_s(\boldsymbol{y}_s, \boldsymbol{\theta}_s)$ as an identity function, *i.e.*, $\boldsymbol{\theta}_s = \mathbf{I}$, and we treat $\boldsymbol{y}_s$ as a score vector on each class.

Table 2 shows the results of ZooBP and HALO on the DBLP-bipartite and Academic datasets as described above. We also report the itemized classification accuracy on "Author" and "Paper" node types. From these results, we observe that HALO outperforms ZooBP in all settings. For large $k$, the gap is generally more significant. The main reason for this phenomenon is that while HALO utilizes both node features and trainable compatibility matrices, ZooBP only uses training labels with fixed compatibility matrices. To the best of our knowledge, HALO is the only method designed to exploit a trainable compatibility matrix.

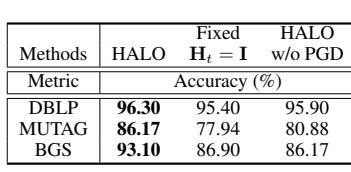

| Methods | HALO | Fixed $\mathbf{H}_t = \mathbf{I}$ | HALO w/o PGD |
|---|---|---|---|
| Metric | Accuracy (%) | | |
| DBLP | **96.30** | 95.40 | 95.90 |
| MUTAG | **86.17** | 77.94 | 80.88 |
| BGS | **93.10** | 86.90 | 86.17 |

Table 3: Ablation study

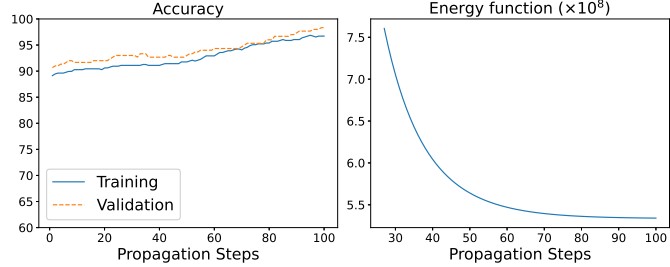

Figure 1: Accuracy and energy function value versus the number of propagation steps on ACM.

## 4.3 Further HALO Analysis

In this section, we first carry out an ablation study to assess the importance of learnable compatibility matrices $\mathbf{H}_t \in \mathcal{H}$, as well as the value of non-linear graph propagation operators instantiated through PGD. Later, we also show the accuracy and energy values of HALO versus the number of propagation steps to demonstrate convergence of the lower-level HALO optimization process, which naturally mitigates oversmoothing.

**Ablation Study** To examine the effectiveness of trainable compatibility matrices, we enforce $\mathbf{H}_t = \mathbf{I}$ for all $t$, using the same embedding dimension across all node types. Other training conditions remain unchanged. Table 3 compares the resulting accuracy on DBLP, MUTAG, and BGS datasets, where we observe that HALO performs significantly better, especially on MUTAG and BGS. These results show that using trainable compatibility matrices to quantify ambiguous relationships between neighboring nodes can be an effective strategy when performing node classification with heterogeneous graphs. Note that even when the embedding dimensions are same (as in these experiments), there can still be

---

[9]The graph in DBLP-reduced is a bipartite graph since the edges exist only between "Paper" and "Author" nodes, while the graph in Academic-reduced also has edges between the "Paper" nodes.

relation-specific mismatches between different node types that exhibit heterophily characteristics, which trainable compatibility matrices can help to resolve.

Turning to the non-linear activations, we also re-run the HALO experiments with (17) removed from the algorithm, meaning PGD defaults to regular SGD as no proximal step is used. As shown in Table 3, HALO without PGD performs considerably worse, implying that the inclusion of the additional non-linear activation leads to better generalization.

**Results Varying Propagation Steps**  Figure 1 shows the results of HALO with various numbers of propagation steps. Specifically, the left plot shows the training and validation accuracy on the ACM dataset, and the right plot shows the energy function value (9) during the unfolding steps. Notably, when we increase the number of propagation steps with a suitable $\alpha$, both training and validation accuracies are gradually increasing, and the energy function value steadily decreasing as expected. These results suggest that as the gradient descent steps gradually decrease the low-level energy function, the corresponding node embeddings $\mathbf{Y}_s$ that emerge are effective in performing the downstream classification task. We have also repeated this experiment using the AIFB dataset and the overall trend is same; for more details please refer to the Supplementary Materials.

# 5    Conclusion

In this paper, we have proposed HALO, a novel HGNN architecture derived from the minimization steps of a relation-aware energy function, the latter consisting of trainable relation-dependent compatibility matrices that can resolve the mismatch between two different node types. Because of intrinsic properties of this unfolding framework, HALO is naturally robust to oversmoothing problems, and outperforms SOTA models on various benchmark datasets. For future work, extending HALO to spatio-temporal graphs in which the node and edge information change continuously can be considered.

# Acknowledgments

This work was supported in part by the New Faculty Startup Fund from Seoul National University, NRF grants [NRF-2021R1A2C2007884, NRF-2021M3E5D2A01024795] and IITP grants [No.2021-0-01343, No.2021-0-02068, No.2022-0-00959, No.2022-0-00113] funded by the Korean government, and SNU-NAVER Hyperscale AI Center. We also thank Zheng Zhang from Amazon Web Services for suggesting interesting connections with techniques for learning knowledge graph embeddings.

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
