# Supplementary Materials for Descent Steps of a Relation-Aware Energy Produce Heterogeneous Graph Neural Networks

**Hongjoon Ahn**[1]*, **Yongyi Yang**[2]*, **Quan Gan**[3], **Taesup Moon**[1]† **and David Wipf**[3]†

[1] ECE/IPAI/ASRI/INMC, Seoul National University, [2] University of Michigan, [3] Amazon Web Services

{hong0805, tsmoon}@snu.ac.kr, yongyi@umich.edu,
quagan@amazon.com, davidwipf@gmail.com

## 1  Proof of Lemma (3.1, Manuscript)

**Lemma 1.1.** *The unique solution* $\mathbf{Y}^*(\mathcal{W}, \mathcal{H})$ *minimizing (9, Manuscript) satisfies*

$$vec(\mathbf{Y}^*(\mathcal{W}, \mathcal{H})) = (\mathbf{I} + \lambda(\mathbf{Q} - \mathbf{P} + \mathbf{D}))^{-1} vec(\widetilde{f}(\widetilde{\mathbf{X}}; \mathcal{W})), \tag{1}$$

*where* $vec(\mathbf{B}) = [\mathbf{b}_1^\top, ..., \mathbf{b}_n^\top]^\top$ *for matrix* $\mathbf{B}$, $\mathcal{H} \triangleq \{\mathbf{H}_t\}_{t \in \mathcal{T}}$ *denotes the set of all compatibility matrices, and we have defined the matrices* $\mathbf{P}$, $\mathbf{Q}$, *and* $\mathbf{D}$ *as*

$$\mathbf{P} = \begin{bmatrix} \mathbf{P}_{11} & ... & \mathbf{P}_{1|\mathcal{S}|} \\ ... & ... & ... \\ \mathbf{P}_{|\mathcal{S}|1} & ... & \mathbf{P}_{|\mathcal{S}||\mathcal{S}|} \end{bmatrix}; \mathbf{P}_{ss'} = \sum_{t \in \mathcal{T}_{ss'}} ((\mathbf{H}_t^\top + \mathbf{H}_{t_{inv}}) \otimes \mathbf{A}_t)$$

$$\mathbf{Q} = \bigoplus_{s \in \mathcal{S}} \mathbf{Q}_s; \mathbf{Q}_s = \sum_{s' \in \mathcal{S}} \sum_{t \in \mathcal{T}_{ss'}} (\mathbf{H}_t \mathbf{H}_t^\top \otimes \mathbf{D}_{st}), \mathbf{D} = \bigoplus_{s \in \mathcal{S}} \mathbf{I} \otimes \mathbf{D}_s.$$

*Here* $\otimes$ *denotes the Kronecker product,* $\bigoplus_{i=1}^n \mathbf{A}_i = diag(\mathbf{A}_1, ..., \mathbf{A}_n)$ *denotes a direct sum of* $n$ *square matrices* $\mathbf{A}_1, ..., \mathbf{A}_n$, *and* $\mathbf{D}_s \triangleq \sum_{s' \in \mathcal{S}} \sum_{t \in \mathcal{T}_{ss'}} \mathbf{D}_{s't}$ *represent a sum of degree matrices over all node types* $s' \in \mathcal{S}$ *and all edge types* $t \in \mathcal{T}_{ss'}$.

Before proceeding the proof of Lemma 1.1, we first provide a basic mathematical result.

**Lemma 1.2.** *(Roth's Column Lemma [1]). For any three matrices* $\mathbf{X}, \mathbf{Y}$ *and* $\mathbf{Z}$,

$$vec(\mathbf{XYZ}) = (\mathbf{Z}^\top \otimes \mathbf{X}) vec(\mathbf{Y}) \tag{2}$$

We now proceed with the proof of our result.

*Proof.* The gradient of (9, Manuscript) is as follows:

$$\nabla_{\mathbf{Y}_s} \ell_{\mathbf{Y}}(\mathbf{Y}) = (\mathbf{I} + \lambda \mathbf{D}_s) \mathbf{Y}_s - f(\mathbf{X}_s; \mathbf{W}_s) + \lambda \sum_{s' \in \mathcal{S}} \sum_{t \in \mathcal{T}_{ss'}} \left( \mathbf{D}_{st} \mathbf{Y}_s (\mathbf{H}_t \mathbf{H}_t^\top) - \mathbf{A}_t \mathbf{Y}_{s'} (\mathbf{H}_t^\top + \mathbf{H}_{t_{inv}}) \right).$$
$$\tag{3}$$

By applying a $vec(\cdot)$ operation to both sides of (3) we obtain:

---

*Work completed during an internship at the AWS Shanghai AI Lab.
†Co-corresponding author.

36th Conference on Neural Information Processing Systems (NeurIPS 2022).

$$vec(\nabla_{\mathbf{Y}_s}\ell(\mathbf{Y})) = vec(\mathbf{Y}_s) - vec(f(\mathbf{X}_s; \mathbf{W}_s))$$
$$+ \lambda(\sum_{s' \in \mathcal{S}} \sum_{t \in \mathcal{T}_{ss'}} (vec(\mathbf{D}_{st}\mathbf{Y}_s(\mathbf{H}_t\mathbf{H}_t^\top)) - vec(\mathbf{A}_t\mathbf{Y}_{s'}(\mathbf{H}_t^\top + \mathbf{H}_{t_{\text{inv}}}))) + vec(\mathbf{D}_s\mathbf{Y}_s))$$

$$(4)$$

Here, using Roth's Column Lemma 1.2, we define the matrices $\mathbf{P}, \mathbf{Q}, \mathbf{D}$ as follows:

$$\mathbf{P} = \begin{bmatrix} \mathbf{P}_{11} & ... & \mathbf{P}_{1|\mathcal{S}|} \\ ... & ... & ... \\ \mathbf{P}_{|\mathcal{S}|1} & ... & \mathbf{P}_{|\mathcal{S}||\mathcal{S}|} \end{bmatrix}; \mathbf{P}_{ss'} = \sum_{t \in \mathcal{T}_{ss'}} ((\mathbf{H}_t^\top + \mathbf{H}_{t_{\text{inv}}}) \otimes \mathbf{A}_t)$$

$$\mathbf{Q} = \bigoplus_{s \in \mathcal{S}} \mathbf{Q}_s; \mathbf{Q}_s = \sum_{s' \in \mathcal{S}} \sum_{t \in \mathcal{T}_{ss'}} (\mathbf{H}_t\mathbf{H}_t^\top \otimes \mathbf{D}_{st}), \mathbf{D} = \bigoplus_{s \in \mathcal{S}} \mathbf{I} \otimes \mathbf{D}_s.$$

Therefore, after rewriting (4) as

$$vec(\nabla_{\mathbf{Y}_s}\ell_{\mathbf{Y}}(\mathbf{Y})) = vec(\mathbf{Y}_s) - vec(f(\mathbf{X}_s; \mathbf{W}_s))$$
$$+ \lambda(\mathbf{Q}_s vec(\mathbf{Y}_s) - \sum_{s' \in \mathcal{S}} \mathbf{P}_{ss'} vec(\mathbf{Y}_{s'}) + \mathbf{D}_s vec(\mathbf{Y}_s)), \quad (5)$$

we can stack $|\mathcal{S}|$ such matrix equations together using $\mathbf{P}, \mathbf{Q}, \mathbf{D}$ to obtain:

$$vec(\nabla_{\mathbf{Y}}\ell_{\mathbf{Y}}(\mathbf{Y})) = vec(\mathbf{Y}) - vec(\widetilde{f}(\widetilde{\mathbf{X}}; \mathcal{W})) + \lambda(\mathbf{Q} - \mathbf{P} + \mathbf{D})vec(\mathbf{Y}). \quad (6)$$

Since $\ell_{\mathbf{Y}}(\mathbf{Y})$ is a convex function, a point that achieves $\nabla_{\mathbf{Y}}\ell_{\mathbf{Y}}(\mathbf{Y}) = 0$ is a optimal point. Therefore, the closed-form solution for $vec(\mathbf{Y}^*(\mathcal{W}, \mathcal{H}))$ is derived as:

$$vec(\mathbf{Y}^*(\mathcal{W}, \mathcal{H})) = (\mathbf{I} + \lambda(\mathbf{Q} - \mathbf{P} + \mathbf{D}))^{-1} vec(\widetilde{f}(\widetilde{\mathbf{X}}; \mathcal{W})). \quad (7)$$

$\square$

## 2   Proof of Theorem (3.2, Manuscript)

**Theorem 2.1.** *The iterations (10) are guaranteed to monotonically converge to the unique global minimum of (9) provided that*

$$\alpha < \frac{2 + 2\lambda d_{min}}{1 + \lambda(d_{min} + \sigma_{max})}, \quad (8)$$

*where $d_{min}$ is the minimum diagonal element of $\mathbf{D}$ and $\sigma_{max}$ is a maximum eigenvalue of matrix $(\mathbf{Q} - \mathbf{P})$.*

*Proof.* The energy function and iteration $k + 1$ of gradient descent on $\mathbf{Y}_s$ with preconditioning is as follows:

$$\ell_{\mathbf{Y}}(\mathbf{Y}) \triangleq \sum_{s \in \mathcal{S}} \left[ \frac{1}{2} ||\mathbf{Y}_s - f(\mathbf{X}_s; \mathbf{W}_s)||_{\mathcal{F}}^2 + \frac{\lambda}{2} \sum_{s' \in \mathcal{S}} \sum_{t \in \mathcal{T}_{ss'}} \sum_{(i,j) \in \mathcal{E}_t} ||\boldsymbol{y}_{si}\mathbf{H}_t - \boldsymbol{y}_{s'j}||_2^2 \right] \quad (9)$$

$$\mathbf{Y}_s^{(k+1)} = \mathbf{Y}_s^{(k)} - \alpha\widetilde{\mathbf{D}}_s^{-1}\nabla_{\mathbf{Y}_s^{(k)}}\ell_{\mathbf{Y}}(\mathbf{Y}). \quad (10)$$

By applying the $vec(\cdot)$ operation to both sides of 10, this can be transformed into:

$$vec(\mathbf{Y}_s^{(k+1)}) = vec(\mathbf{Y}_s^{(k)}) - \alpha vec(\widetilde{\mathbf{D}}_s^{-1}\nabla_{\mathbf{Y}_s^{(k)}}\ell_{\mathbf{Y}}(\mathbf{Y})) \tag{11}$$

$$= vec(\mathbf{Y}_s^{(k)}) - \alpha(\mathbf{I} \otimes \widetilde{\mathbf{D}}_s^{-1})vec(\nabla_{\mathbf{Y}_s^{(k)}}\ell_{\mathbf{Y}}(\mathbf{Y})). \tag{12}$$

Note that we apply Roth's column lemma to (11) to derive (12). Stacking $|\mathcal{S}|$ such vectors together, (12) can be written as:

$$vec(\mathbf{Y}^{(k+1)}) = vec(\mathbf{Y}^{(k)}) - \alpha\widetilde{\mathbf{D}}^{-1}vec(\nabla_{\mathbf{Y}^{(k)}}\ell_{\mathbf{Y}}(\mathbf{Y})), \tag{13}$$

where $\widetilde{\mathbf{D}}^{-1} \triangleq \bigoplus_{s\in\mathcal{S}} \mathbf{I} \otimes \widetilde{\mathbf{D}}_s^{-1}$.

Because $\ell_{\mathbf{Y}}(\mathbf{Y})$ is convex, for any $\mathbf{Y}^{(k+1)}$ and $\mathbf{Y}^{(k)}$ the following inequality holds:

$$\ell_{\mathbf{Y}}(\mathbf{Y}^{(k+1)}) \le \ell_{\mathbf{Y}}(\mathbf{Y}^{(k)}) + vec(\nabla_{\mathbf{Y}^{(k)}}\ell_{\mathbf{Y}}(\mathbf{Y}))^\top vec(\mathbf{Y}^{(k+1)} - \mathbf{Y}^{(k)})$$
$$+ \frac{1}{2}vec(\mathbf{Y}^{(k+1)} - \mathbf{Y}^{(k)})^\top \nabla^2_{\mathbf{Y}^{(k)}}\ell_{\mathbf{Y}}(\mathbf{Y})vec(\mathbf{Y}^{(k+1)} - \mathbf{Y}^{(k)}), \quad (14)$$

where $\nabla^2_{\mathbf{Y}^{(k)}}\ell_{\mathbf{Y}}(\mathbf{Y})$ is a Hessian matrix whose elements are $\nabla^2_{\mathbf{Y}^{(k)}}\ell_{\mathbf{Y}}(\mathbf{Y})_{ij} = \frac{\partial\ell_{\mathbf{Y}}(\mathbf{Y})}{\partial vec(\mathbf{Y})_i \partial vec(\mathbf{Y})_j}\big|_{\mathbf{Y}=\mathbf{Y}^{(k)}}$.

Plugging in the gradient descent update by letting $vec(\mathbf{Y}^{(k+1)} - \mathbf{Y}^{(k)}) = -\alpha\widetilde{\mathbf{D}}^{-1}vec(\nabla_{\mathbf{Y}^{(k)}}\ell_{\mathbf{Y}}(\mathbf{Y}))$, we get:

$$\ell_{\mathbf{Y}}(\mathbf{Y}^{(k+1)}) \le \ell_{\mathbf{Y}}(\mathbf{Y}^{(k)}) - (\widetilde{\mathbf{D}}^{-1}vec(\nabla_{\mathbf{Y}^{(k)}}\ell_{\mathbf{Y}}(\mathbf{Y})))^\top(\alpha\widetilde{\mathbf{D}})(\widetilde{\mathbf{D}}^{-1}vec(\nabla_{\mathbf{Y}^{(k)}}\ell_{\mathbf{Y}}(\mathbf{Y})))$$
$$+ (\widetilde{\mathbf{D}}^{-1}vec(\nabla_{\mathbf{Y}^{(k)}}\ell_{\mathbf{Y}}(\mathbf{Y})))^\top(\frac{\alpha^2}{2}\nabla^2_{\mathbf{Y}^{(k)}}\ell_{\mathbf{Y}}(\mathbf{Y}))(\widetilde{\mathbf{D}}^{-1}vec(\nabla_{\mathbf{Y}^{(k)}}\ell_{\mathbf{Y}}(\mathbf{Y}))). \tag{15}$$

If $\alpha\widetilde{\mathbf{D}} - \frac{\alpha^2}{2}\nabla^2_{\mathbf{Y}^{(k)}}\ell_{\mathbf{Y}}(\mathbf{Y}) \succ 0$ holds, then gradient descent will never increase the loss, and moreover, since $\ell_{\mathbf{Y}}(\mathbf{Y})$ is strongly convex, it will monotonically decrease the loss until the unique global minimum is obtained. To compute $\nabla^2_{\mathbf{Y}^{(k)}}\ell_{\mathbf{Y}}(\mathbf{Y})$, we differentiate (6) and arrive at:

$$\nabla^2_{\mathbf{Y}^{(k)}}\ell_{\mathbf{Y}}(\mathbf{Y}) = \mathbf{I} + \lambda(\mathbf{Q} - \mathbf{P} + \mathbf{D}). \tag{16}$$

Returning to the above inequality, we can then proceed as follows:

$$\alpha\widetilde{\mathbf{D}} - \frac{\alpha^2}{2}(\mathbf{I} + \lambda(\mathbf{Q} - \mathbf{P} + \mathbf{D})) = \alpha(\mathbf{I} + \lambda\mathbf{D}) - \frac{\alpha^2}{2}(\mathbf{I} + \lambda(\mathbf{Q} - \mathbf{P} + \mathbf{D}))$$
$$= (\alpha - \frac{\alpha^2}{2})(\mathbf{I} + \lambda\mathbf{D}) - \frac{\alpha^2\lambda}{2}(\mathbf{Q} - \mathbf{P})$$
$$\succ (\alpha - \frac{\alpha^2}{2})(1 + \lambda d_{\min})\mathbf{I} - \frac{\alpha^2\lambda}{2}(\mathbf{Q} - \mathbf{P}). \tag{17}$$

If $\alpha$ satisfies $(\alpha - \frac{\alpha^2}{2})(1 + \lambda d_{\min})\mathbf{I} - \frac{\alpha^2\lambda}{2}(\mathbf{Q} - \mathbf{P}) \succ 0$, then $\alpha\widetilde{\mathbf{D}} - \frac{\alpha^2}{2}\nabla^2_{\mathbf{Y}^{(k)}}\ell_{\mathbf{Y}}(\mathbf{Y}) \succ 0$ holds. Therefore, a sufficient condition for convergence to the unique global optimum is:

$$(\alpha - \frac{\alpha^2}{2})(1 + \lambda d_{\min}) - \frac{\alpha^2\lambda}{2}\sigma_{\max} > 0. \tag{18}$$

Consequently, to guarantee the aforementioned convergence we arrive at the final inequality:

$$\alpha < \frac{2 + 2\lambda d_{\min}}{1 + \lambda(d_{\min} + \sigma_{\min})}. \tag{19}$$

$\square$

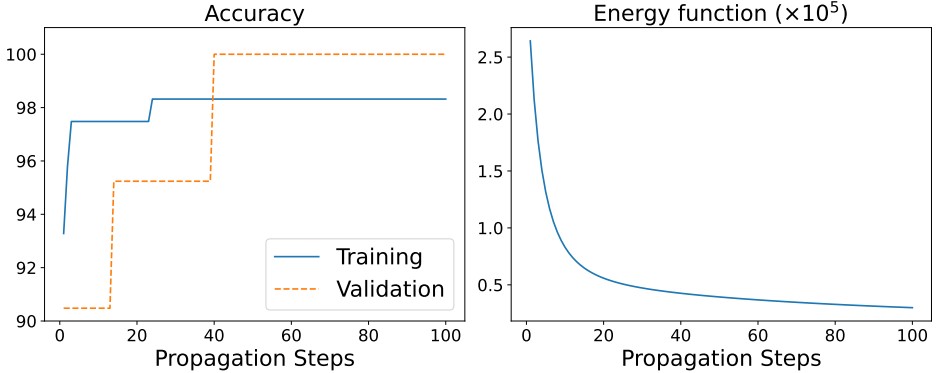

Figure 1: Accuracy and energy function value versus the number of propagation steps on AIFB.

## 3 Additional Experiment results

### 3.1 Results Varying Propagation Steps

Figure 1 shows the results of HALO with various number of propagation steps in AIFB dataset experiment. Similar to the results on the ACM dataset in our main paper, the training and validation accuracy increase when we increase the number of propagation steps, while the energy function value is steadily decreasing.

### 3.2 Results on Different Base Models and Test Time Comparison

Table 1 shows the results of applying HALO with different base models, as well as test time comparison between HALO and R-GCN[3]. For the former, we use a 2-layer MLP $f(\mathbf{X}; \mathbf{W}_s)$ per node type. In Table 1 (left), HALO with the MLP base model achieves comparable results to HALO using a linear layer for the base model. Of course with larger or more complex datasets, an MLP could potentially still be advantageous.

As we mentioned in the main paper, the time complexity of propagating the graph features in HALO is on the same level as R-GCN. To corroborate this claim empirically, we carry out an experiment comparing the test times of HALO and R-GCN. In this experiment, both R-GCN and HALO used 16 GNN layers with 16 hidden dimensions. In Table 1 (right), as expected, R-GCN and HALO take almost same amount of time when the model sizes are same.

Table 1: Results using different base models (left) and test time comparisons (right).

| Datasets | MUTAG | BGS | DBLP |
| --- | --- | --- | --- |
| Metric | Accuracy (%) | | |
| HALO | 86.17 | 93.10 | 96.30 |
| HALO w/ MLP $f(\mathbf{X}; \mathbf{W}_s)$ | 85.58 | 93.79 | 94.75 |

| Datasets | MUTAG | AIFB |
| --- | --- | --- |
| Metric | Time (second) | |
| R-GCN | 0.8424 | 1.8403 |
| HALO | 0.7740 | 1.7065 |

### 3.3 Results with standard error

Table 2 reproduces the results from Table 1 in the main paper, but with the average accuracy and corresponding standard deviations across 5 runs included.

## 4 Details on Experiment Settings

### 4.1 Hyperparameters for the experimental results from Table 1 (Manuscript)

In the all experiments, we used Adam optimizer [2] and dropout as regularization with dropout rate 0.5. For other hyperparameters, please refer to Table 3.

Table 2: Results on HGB (left) and knowledge graphs (right). The results are averaged over 5 runs, with standard deviations included.

| Dataset | DBLP | IMDB | ACM | Freebase |
|---|---|---|---|---|
| Metric | Accuracy (%) | | | |
| R-GCN | 92.07± 0.50 | 62.05± 0.15 | 91.41± 0.75 | 58.33± 1.57 |
| HAN | 92.05± 0.62 | 64.63± 0.58 | 90.79± 0.43 | 54.77± 1.40 |
| HGT | 93.49± 0.25 | 67.20± 0.57 | 91.00± 0.76 | 60.51± 1.16 |
| Simple-HGN | 94.46±0.22 | 67.36±0.57 | 93.35±0.45 | **66.29±0.45** |
| HALO | **96.30±0.46** | **76.20±0.77** | **94.33±1.00** | 62.06±0.74 |

| Dataset | AIFB | MUTAG | BGS | AM |
|---|---|---|---|---|
| Metric | Accuracy (%) | | | |
| Feat | 55.55±0.00 | 77.94±0.00 | 72.41±0.00 | 66.66±0.00 |
| WL | 80.55±0.00 | 80.88±0.00 | 86.20±0.00 | 87.37±0.00 |
| RDF2Vec | 88.88±0.00 | 67.20±1.24 | 87.24±0.89 | 88.33±0.61 |
| R-GCN | 95.83±0.62 | 73.23±0.48 | 83.10±0.80 | 89.29±0.35 |
| HALO | **96.11±2.22** | **86.17±1.18** | **93.10±2.18** | **90.20±1.20** |

Table 3: Model Hyperparameters for HALO

| Datasets | DBLP | IMDB | ACM | Freebase | AIFB | MUTAG | BGS | AM |
|---|---|---|---|---|---|---|---|---|
| Hidden Layer Size | 256 | 64 | 32 | 32 | 16 | 16 | 16 | 16 |
| Learning Rate | $10^{-4}$ | $10^{-3}$ | $10^{-2}$ | $10^{-2}$ | $10^{-3}$ | $10^{-3}$ | $10^{-2}$ | $10^{-2}$ |
| Weight Decay | $10^{-5}$ | $10^{-5}$ | $10^{-4}$ | $10^{-3}$ | $10^{-5}$ | $10^{-4}$ | $10^{-5}$ | $10^{-4}$ |
| K | 8 | 32 | 32 | 4 | 16 | 16 | 8 | 4 |
| $\lambda$ | 1 | 1 | 0.1 | 1 | 1 | 0.01 | 0.1 | 1 |
| $\alpha$ | 1 | 1 | 0.1 | 1 | 0.1 | 1 | 1 | 1 |

## 4.2 Categorization in DBLP and Academic datasets

To carry out experiments for ZooBP, we modified DBLP and Academic datasets as mentioned in the main text. Here, we provide details on the categorization of venues in these datasets. Each venue is the name of academic conference.

**DBLP** dataset has 20 venues: AAAI, CVPR, ECML, ICML, IJCAI, SIGMOD, VLDB, EDBT, ICDE, PODS, ICDM, KDD, PAKDD, PKDD, SDM, CIKM, CIR, SIGIR, WSDM, WWW. We categorize the above venues to 4 categories: **ML** (AAAI, CVPR, ECML, ICML, IJCAI), **DB** (SIGMOD, VLDB, EDBT, ICDE, PODS), **DM** (ICDM, KDD, PAKDD, PKDD, SDM), and **IR**(CIKM, CIR, SIGIR, WSDM, WWW).

**Academic** dataset has 18 venues: ICML, AAAI, IJCAI, CVPR, ICCV, ECCV, ACL, EMNLP, NAACL, KDD, WSDM, ICDM, SIGMOD, VLDB, ICDE, WWW, SIGIR, CIKM. We categorize the above venues to 4 categories: **ML** (ICML, AAAI, IJCAI), **Vision** (CVPR, ICCV, ECCV), **NLP** (ACL, EMNLP, NAACL), and **Data** (KDD, WSDM, ICDM, SIGMOD, VLDB, ICDE, WWW, SIGIR, CIKM).

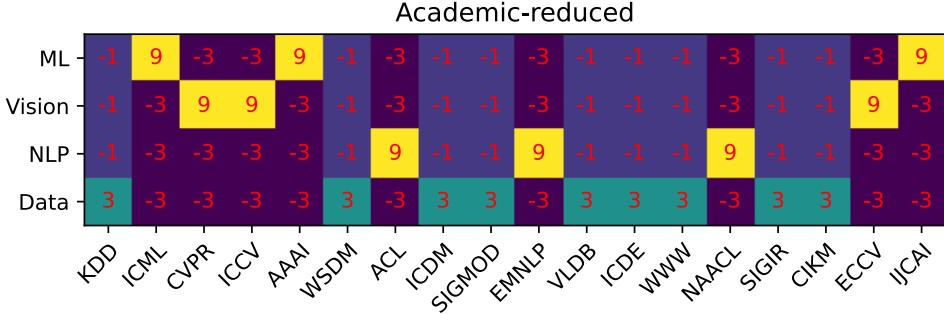

Figure 2: Compatibility matrix used in Academic-reduced datsaet

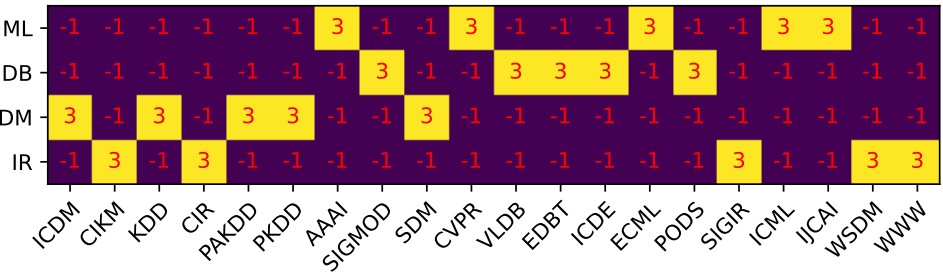

Figure 3: Compatibility matrix used in DBLP-reduced datsaet

For choosing the compatibility matrix of ZooBP, based on the category of each venue, we give high positive value on $H(i,j)$ if $j$-th venue corresponds to $i$-th venue. For example, "AAAI" in DBLP dataset belongs to "ML" category. Therefore, we give high positive value on $H(\text{"ML"}, \text{"AAAI"})$. Otherwise, we give negative value to satisfy the residual condition of the compatibility matrix. The results for the Academic and DBLP data are shown in Figures 2 and 3 respectively.

## 5    Limitations and Potential Negative Social Impact

One limitation is that we have thus far not integrated HALO with large-scale sampling, which would allow us to apply it to huge graphs. And a potential negative societal impact is that as the node classification accuracy with heterogeneous graphs significantly improves with models like HALO, it could be maliciously used to improve the quality of the recommendation of socially damaging products on the Internet (e.g., dangerous weapons or harmful videos on streaming websites).