# OpenReview forum: "Descent Steps of a Relation-Aware Energy Produce Heterogeneous Graph Neural Networks"
_NeurIPS.cc/2022/Conference — NeurIPS 2022 Accept_

### Official Review · Reviewer_U2aH · 2022-06-22

**Rating:** 5
**Confidence:** 4
**Soundness:** 3 good
**Presentation:** 3 good
**Contribution:** 3 good

**Summary:**

This paper studies heterogeneous GNN and proposes a novel HGNN architecture HALO, derived from the minimization steps of a relation-aware energy function. HALO introduces trainable compatibility matrices to solve the mis-matches between two different node types. Experiments on 8 benchmark datasets show the effectiveness of the proposed model.

**Questions:**

Please see the comments above.

**Limitations:**

I think the high space complexity of the model is a limitation that needs to be further improved, which is omitted by the authors.

**Strengths And Weaknesses:**

Strength:

1 The paper is well-written and easy to follow.

2 The proposed method HALO is derived by the gradient descent steps for the minimization of a relation-aware energy function, which is more explainable.

3 Experiments on 8 benchmark datasets show the superiority of the proposed method.

Weakness:

1. The proposed method HALO is a direct extension to the model proposed in [32] to heterogeneous graphs. Compared with the model in [32], HALO introduces compatibility matrices to map embeddings of nodes of different types into the same low-dimensional space. Therefore, the proposed model is a bit incremental.
2. There exist some confusions in the paper. For example, in line 133, the authors write “Eq 5 fail to capture the nuances of heterogeneous data”. I think this is incorrect because in Eq 5, node types are taken as nodes features, and edges types are also considered.
3. For experiments, I have several concerns. First, for Table 1, I even do not know what the evaluation measure is before I step into the main texts. Further, based on one metric of classification accuracy, the results is somewhat unconvincing. Finally, since alleviating the over-smoothing issue is an important contribution of the proposed model, conducting experiments on more datasets is a necessity, while in the current version, the only results on ACM seem to be cherry-pick.
4. Some important baselines are missing in this area:
(1) Representation learning for attributed multiplex heterogeneous network
(2) Graph Transformer Networks
(3) Leveraging Meta-path Contexts for Classification in Heterogeneous Information Networks
5. While 8.9 pages are not a big issue, the blank part is very easy to fill, which can make the paper more compact.

---

> ### Author Response · Authors · 2022-08-02
> **Response to Reviewer U2aH (Part I)**
>
> Thanks for the constructive feedback and positive comments regarding the clarity, explainability and superiority of our proposed approach.  We address unresolved questions/issues below.
>
> ### Weakness 1:  The proposed model can be viewed as an incremental extension of the homogeneous model from reference [32]
>
> There are actually many unfolded-optimization-based methods applied to *homogeneous* graphs (e.g., references [2,16,20,32,33,35] mentioned in our submission); however, none of these address general heterogeneous extensions because it is *a priori* not at all obvious how to do so (see Section 3.1 for some naive attempts).  Of course retrospectively we can always examine our approach and the connections with prior homogeneous models, but this does not undercut the novelty involved in handling label and distribution mismatches across different node/edge types.  In fact, most heterogeneous GNNs bear some similarity to a prior/simpler homogeneous counterpart, e.g., GCN vs. RGCN, etc.  Given this context then, we feel that our proposed energy function and convergent descent iterations form a non-trivial contribution to the community, especially given that many/most real-world graphs tend to be heterogeneous.
>
> ### Weakness 2:  Statement regarding Eq. (5) on line 133 may be incorrect
>
> We believe the statement regarding Eq. (5) on line 133 is correct; however, we provide some context that might help clarify the intended message.  We can also add such clarification to a revised/final version, as others may have a similar concern. Thanks for pointing out that this statement might not be adequately clear.
>
> To begin, Eq. (5) (and equivalently Eq. (6)) is composed of two terms.  While the first term can be tailored to different node types, it is invariant to different edge/relation types.  In contrast, our comment on line 133 is primarily directed towards the second, edge-dependent term, which from Eq. (6) reduces to a penalty given by $\mathrm{tr}(Y^T C Y)$, where $C$ is a PSD matrix formed by summing across relations and node types.  This summation completely blurs out the nuances of each edge type, and penalties of this general quadratic, PSD form are actually a commonly used regularization factor for *homogeneous* graphs (see reference [11] in our paper).
>
> Quite differently, our HALO model includes trainable parameters for each relation type that cannot be collapsed or aggregated into a simple low-dimensional quadratic factor such as $C$.  In this way, different edge types can receive quite different regularization effects as needed for downstream predictions.
>
> ### Weakness 3: Concerns regarding experiments
> With respect to evaluation measures, we will add these to table captions in the revision (as opposed to just embedding in the text which is admittedly harder to find).  Good suggestion, we should have caught this in the original submission.
>
> In terms of the accuracy results themselves, we believe that our model does produce significant improvement over existing baselines.  For example, in Table 1 HALO achieves roughly 2% average improvement over the next best model on HGB datasets, and 6% average improvement on the knowledge graphs.  These differences are sizeable given the margins often displayed in GNN papers.  Similarly, HALO outperforms ZooBP by even much wider margins in Table 2.
>
> Finally, with respect to the evaluation of oversmoothing in Figure 1, the basic trend is not restricted to the ACM dataset.  This is perhaps not surprising given that, (i) with suitably-small step-size the proposed unfolded optimization process is guaranteed to converge, and (ii) provided $\lambda$ is chosen such that the minimum energy of Eq. (8) is not oversmoothed, then executing an arbitrary number of steps/layers will closely approximate this minimum and cannot possibly oversmooth.  Even so, we agree with the reviewer that more examples may help to illustrate this point.  Hence we have repeated the process used to create Figure 1 with the AIFB dataset as well, and the trend is the same as expected.  Thanks for the suggestion; we will add these new figures to a final/revised version of the paper, space permitting (or else the supplementary if not).

---

> > ### Author Response · Authors · 2022-08-02
> > **Response to Reviewer U2aH  (Part II)**
> >
> > ### Weakness 4: Missing baselines
> > Thank you for recommending additional references to consider. We can add these papers to our related work section to provide broader perspective. However, we do not include empirical comparisons with them for the following reasons: 1) We already compare against more recent state-of-the-art (SOTA) HGNN methods as listed and benchmarked in reference [15].  In this regard, we believe that our comparisons in Table 1 are sufficient relative to SOTA across standardized benchmark settings. 2) To the best of our knowledge, all of the suggested methods require meta-paths manually-defined w.r.t. each individual heterogeneous graph.  Although this can be useful in some circumstances, it is quite unlike HALO, which can automatically learn capatability matrices for each dataset without such user involvement.  (Note also that some of the suggested references rely on non-standardized experimental designs that are not comparable to ours.)
> >
> > ### Limitations:  High space complexity may be a limitation of the proposed method
> >
> > Just to clarify, the space complexity of HALO is essentially the same as the commonly-used RGCN heterogeneous GNN model, and is not actually a primary concern or limitation.  We emphasize that the large matrices $P$ and $Q$ mentioned in Lemma 3.1 are included for analysis and intuition purposes, while the actual implementation uses Eq. (12), whose space complexity matches RGCN.

---

> ### Comment · Reviewer_U2aH · 2022-08-05
> **Conclusion**
>
> I have carefully read all the reviews and the rebuttal letter. Thanks for the clarification. However, I have to restate that all the technique details in this paper including gradient descent based optimization and related proofs are directly extended from [32], which makes the paper a bit incremental. So I will keep my score.

---

> > ### Author Response · Authors · 2022-08-05
> > **Response to the conclusion**
> >
> > We completely agree with the reviewer that the unfolded gradient-based optimization framework comes from prior work; indeed we cite reference [32] five times as well as many other related unfolding-based GNN papers.  However, we reiterate that none of these prior works contain any hint of how to extend to heterogeneous graphs, and the setup and execution for doing so is not at all obvious or incremental from our viewpoint.  That being said, we concede that good-faith differences of opinion can occur, and we nonetheless appreciate the reviewer's willingness to read through our rebuttal and consider the arguments we have presented.

---

### Official Review · Reviewer_Dn37 · 2022-07-11

**Rating:** 6
**Confidence:** 5
**Soundness:** 3 good
**Presentation:** 4 excellent
**Contribution:** 3 good

**Summary:**

This paper propose a new heterogeneous GNN motivated by minimizing relation-aware energy function. In short, it tries to make connected nodes share similar labels, with a relation-dependent compatibility matrices to resolve the mistmatch between different node types. The proposed method achieves improvement on heteogenous graph benchmarks.

**Questions:**

1. The energy regularization seems to be similar to label propagation, and recently there exist several attempts to combine label propagation with GNN, such as [1]. Could the authors discuss their difference?

2. After reading the paper and Alg 1. I think the proposed method is more like a learning framework, and it should be agnostic to different base HGNN architecture (f in alg1)? Could the authors provide more details about it? If it's indeed model-agnostic, could the authors conduct experiments with different base function f?

3. I'm still not pretty clear why the authors choose proximal gradient descent instead of standard optimization. Could the authors provide more discussion or add some ablation study?

[1] Combining Label Propagation and Simple Models Out-performs Graph Neural Networks

**Ethics Review Area:**

["I don’t know"]

**Limitations:**

The authors claim that limitation is included in supplementary but I didn't find it. I suggest the authors to add some discussion about the usage boundary and potential limitation of this method.

**Strengths And Weaknesses:**

Strength:

1. This paper took non-trivial efforts to adopt unfolded optimization techniques (somehow force connected nodes to share similar labels) into heterogeneous GNN. There exist some challenges, such as the label and distribution mismatch across different node types, how to integrate the deterministic label adjustment with the non-linear model (the authors adopt a proximal gradient descent)

2. I appreciate how the authors try to improve the simple form in eq(7) by considering heterophily (not simply considering connected nodes to have similar labels), introducing a projection matrix H, and give a detailed and closed-form solution for the modified method.

Weakness:

1. The proposed method has many design choices, such as a naive regularization in eq (7), standard optimization, etc. Current ablation studies only change projection matrix H. It would be better if the authors could conduct more thorough ablation studies and tell us which component contributes most.

2. This paper has already conducted lots of experiments. However, it would still be good if they could also choose larger datasets such as OGB-MAG and OGBLSC-MAG and report the results. Also, it would be good to also show the model parameter and inference time (or FLOPS).



Personally I like this paper, and I think the paper could be improved if the authors could make the experiments more solid and provide more details and discussion.

---

> ### Author Response · Authors · 2022-08-02
> **Response to Reviewer Dn37 (Part I)**
>
> Thanks for the constructive comments and pointing out the non-trivial contribution of our work in handling label/distribution mismatch across different node types, etc.  We address each reviewer question/issue below.
>
>
> ### Weakness 1: Additional ablation studies needed to further assess design choices
>
> We have carried out additional ablation studies on our method. All the results are presented in the replies to Questions 2 and 3 below.  These can also be later included in a final/revised version of the paper.
>
>
> ### Weakness 2: Comparisons on larger dataset OGB-MAG (or OGBLSC-MAG) and inference time comparisons
>
> While we agree scalability is an important issue, OGB-MAG is such a large dataset that the vast majority of GNN papers do not actually include it for comparison purposes.  In fact, as with other GNN models, to train our approach on OGB-MAG would require sampling and other non-trivial engineering efforts and tuning that are beyond the scope of our submission (and unfortunately, we do not presently have the time and resources to complete this during the rebuttal period).  However, we can consider this for future work, as it would be very interesting to consider impactful theoretical and algorithmic aspects related to the integration of HALO with large-scale sampling.
>
>
> In terms of potential for scalability, as discussed in our submission, the time complexity of HALO is of the same order as RGCN. To further explore empirically, we compare the inference time between HALO and RGCN. We carry out experiments using 16 GNN layers with 16 hidden dimensions; results of HALO and RGCN are shown below, which are consistent with expectations that the inference times are similar when the model sizes are the same.
>
> |Method / Datasets|MUTAG|AIFB|
> |:--------:|:---:|:---:|
> |Time (second)|||
> |R-GCN|0.8424|1.8403|
> |HALO|0.7740|1.7065|
>
> ### Question 1: Discussion of differences from prior efforts to combine label propagation methods with GNNs
>
> The reference the reviewer mentions ("Combining label propagtion and simple models out-performs graph neural networks") introduces a 2-stage homogeneous model whereby the first stage simply trains a node-wise base model (i.e., no graph propagation), and the second non-trainable stage smooths the predictions from the first stage using fixed label-propagation-like smoothing operators.  These two stages are not trained end-to-end, nor do they directly apply to heterogeneous graphs.  The latter would represent a non-trivial adaptation, since new heterogeneous node- and edge-specific smoothing operators would need to be defined and tuned.  In contrast, our model involves the end-to-end learning of a node-specific base model combined with trainable node- and edge-specific smoothing operators instantiated through the compatibility matrices $H_t$.
>
>
> ### Question 2: More discussion and experiments with different base models
>
> Indeed, our model is flexible w.r.t. the base model $f$ used in Eq. (8). In our main paper, we choose $f$ as a linear layer to keep the model simple, especially since additional parameters are introduced via the trainable compatibility matrices $H_t$. However, as shown in the ablation results below, our model can effectively adopt an MLP for $f$ as well, and the results are mostly comparable. (For this purpose, we searched the number of MLP hidden layers, and a single hidden layer achieved the best results.)
>
> |Method / Datasets|MUTAG|BGS|DBLP|
> |:--------:|:---:|:---:|:---:|
> |Accuracy (%)|
> |HALO|86.17|93.10|96.30|
> |HALO w/ MLP f(X;W)|85.58|93.79|94.75|
>
> ### Question 3: Proximal gradient descent versus standard optimization
>
> Using proximal gradient descent (PGD) is invaluable for handling non-smooth objective functions that may have flat or non-existent derivatives in regions of parameter space that can disrupt the convergence of standard SGD.  And PDG with the appropriate choice of non-smooth penalty functions can reproduce common non-linear activitions such as ReLU.  To more explicitly demonstrate the effectiveness of PGD in the present context, we carry out an additional ablation experiment whereby the non-smooth penalty and attendant ReLU activation are removed, and regular SGD is applied.  The following table shows the results, where we observe that the performance is significantly degraded.
>
> |Method / Datasets|MUTAG|BGS|DBLP|
> |:--------:|:---:|:---:|:---:|
> |Accuracy (%)||
> |HALO |86.17|93.10|96.30|
> |HALO w/o PGD (w/o ReLU)|80.88|79.30|95.90|

---

> > ### Author Response · Authors · 2022-08-02
> > **Response to Reviewer Dn37 (Part II)**
> >
> > ### Limitations:  Missing discussion of potential limitations and usage boundaries
> >
> > One limitation is that we have thus far not integrated HALO with large-scale sampling, which would allow us to apply it to huge graphs.  A potential negative societal impact is that as the node classication accuracy with heterogeneous graphs significantly improves with models like HALO, it could be maliciously used to improve the quality of the recommendation of socially damaging products on the Internet (e.g., dangerous weapons or harmful videos on streaming websites).  We can add this discussion to a revision (or supplementary if insufficient space).

---

> > > ### Comment · Reviewer_Dn37 · 2022-08-03
> > > **Feedback to Authors' responses**
> > >
> > > Thanks to the authors for conducting many new ablation studies in such a short rebuttal time.
> > >
> > > I highly suggest the authors could provide more details for these experiments, and add into the paper (or at least the appendix), especially the PGD part.
> > >
> > > Also another quick question, for the time experiment, is this time for the inference time of a single batch?
> > >
> > > and for question 2 about base model of f, what I understood is that f could be any model (including GNN). Is it correct? If yes, is it possible that authors could compare choosing one of existing GNN model as f, and compare whether adding the regularization framework proposed in this paper could improve the performance. (from my perspective, the proposed approach could be more useful if it's a general building block that could be added to any model)

---

> > > > ### Author Response · Authors · 2022-08-05
> > > > **Response to the feedback**
> > > >
> > > > Thanks for reading our rebuttal material and the quick follow-up feedback.
> > > >
> > > > ### Details from new experiments
> > > >
> > > > All the settings for the new ablation experiments are same as presented in our original submission, with the exception of the ablated components described as follows:
> > > >
> > > > 1. Base model ablation study: We compare $f$ as a linear layer and as an MLP with different numbers of hidden layers, with dimension equal to the linear model.  Also, for the MLPs, a single hidden layer and ReLU activation performed best in all the new experiments, so we only report this result for the MLPs.
> > > >
> > > > 2. PGD vs. SGD ablation study: When we remove the non-smooth penalty function $\phi$ from the HALO energy (see Eq. (14)), we can minimize the reduced loss with regular SGD instead of PGD.  In practice, this amounts to simply removing the non-linear activations in the unfolding layer, which is equivalent to skipping the proximal step from Eq. (17) during the forward pass; what remains is then just SGD as in Eq. (16) and instantiated via Eq. (12) for HALO.  Except for this, all the hyperparameter sets are same as our original submission.
> > > >
> > > > For reference, we also summarize the hyperparameters involved in the ablations within the following table.
> > > >
> > > > |Methods / Hyperparameters|Hidden dimensions|lr|Weight decay|# GNN layers|f(X;W)|$\lambda$|$\alpha$|
> > > > |:--------:|:---:|:---:|:---:|:---:|:---:|:---:|:---:|
> > > > |MUTAG|
> > > > |HALO|16|0.001|0.0001|16|Linear|0.01|1.0|
> > > > |HALO w/ MLP f(X;W)|16|0.001|0.0001|16|MLP|0.01|1.0|
> > > > |HALO w/o PGD (w/o ReLU)|16|0.001|0.001|16|Linear|0.1|1.0|
> > > > |BGS|
> > > > |HALO|16|0.01|0.00001|8|Linear|0.1|1.0|
> > > > |HALO w/ MLP f(X;W)|16|0.01|0.0001|8|MLP|0.1|1.0|
> > > > |HALO w/o PGD (w/o ReLU)|16|0.01|0.0001|16|Linear|10.0|1.0|
> > > > |DBLP|
> > > > |HALO|256|0.0001|0.00001|8|Linear|1.0|1.0|
> > > > |HALO w/ MLP f(X;W)|256|0.001|0.01|8|MLP|1.0|1.0|
> > > > |HALO w/o PGD (w/o ReLU)|256|0.001|0.00001|8|Linear|0.1|1.0|
> > > >
> > > >
> > > > Note also that once the discussion period has concluded, we will definitely aggregate all reviewer comments and our responses and integrate into a new draft, either the main paper where possible/feasible, or else the supplementary for less critical content that does not fit in the main paper.
> > > >
> > > > ### Follow-up question about inference times
> > > >
> > > > For the inferenece time comparison experiment, we used full-batch inference.
> > > >
> > > > ### Can GNNs be used for base model $f$
> > > > As the reviewer suggests, a GNN could also be used for the base model $f$, and indeed, HALO can be treated as a building block to apply on top of any differentiable base predictor.  That being said, since HALO already involves a trainable/flexible edge/node-type regularization mechanism that modulates the base predictor, it is not clear that addititional message-passing *within* the base predictor itself (as would occur when incorporating a GNN) is necessary to improve performance.

---

> > > > > ### Comment · Reviewer_Dn37 · 2022-08-07
> > > > > **Response to the updated experiments**
> > > > >
> > > > > Thanks for the authors' responses. (The reason for asking replacing f with GNN is just as an option of ablation study to understand the proposed method.) Expect to see these added experiments to be included in the paper.
> > > > >
> > > > > I'll keep the score and support to accept this paper.

---

> > > > > > ### Author Response · Authors · 2022-08-07
> > > > > > **Response to the comments**
> > > > > >
> > > > > > Thanks for the continued engagement with our work during the discussion period. And in terms of our experiments, NeurIPS extends the page limit to 10 pages for final versions, so we would certainly be able to include the additional ablations.

---

### Official Review · Reviewer_Sdud · 2022-07-13

**Rating:** 6
**Confidence:** 3
**Soundness:** 3 good
**Presentation:** 3 good
**Contribution:** 3 good

**Summary:**

This work proposes a novel graph neural network architecture, HALO, for heterogeneous graphs. HALO is derived from the minimization steps of a relation-aware energy function, which addresses the oversmoothing issue in terms of heterogeneous graphs. Extensive experiments conducted on the node classification task in 8 heterogeneous graph datasets show the effectiveness of the proposed method.

**Questions:**

See the Strengths and Weaknesses.

**Limitations:**

The authors didn't address the limitations and potential negative societal impact of their work.

**Strengths And Weaknesses:**

Strengths
1. The approach to solve the oversmoothing issue in terms of heterogeneous graphs is novel.
2. Paper is well-written and easy to follow.
3. The proposed model significantly outperforms other HGNN baselines in several datasets (e.g., IMDB) for node classification.

Weaknesses
1. It would be good if there was a comparison with other works that addressed the oversmoothig issue in homogeneous graphs. Then, the oversmoothing issue in heterogeneous graphs can be experimentally understood more.

---

> ### Author Response · Authors · 2022-08-02
> **Response to Reviewer Sdud**
>
> Thanks for the constructive comments and the appreciation for the novelty, clarity, and effectiveness of our approach. We address each question/issue below.
>
> ### Weakness 1:  Good to add comparison with prior work that addressed the oversmoothing issue in homogeneous graphs
>
> In our ablation study we consider the simplification $H_t=I$ for all $t$, which effectively corresponds to using an energy function that ignores relation-aware regularization and is therefore suitable for homogeneous graphs while being robust to oversmoothing.  And as shown in Table 3 of our submission, the full HALO model (with trained $H_t$) performs considerably better.  Beyond this type of simplification, it is difficult to fairly compare with alternative homogeneous methods that address oversmoothing because such methods do not generally apply to heterogeneous graphs with disparate node and edge types.
>
>
> ### Limitation: Missing discussion of limitations and potential negative societal impact
>
> Thanks for the suggestion.  Actually, we inadvertantly did not include an explicit statement about the limitations and potential negative societal impact of our work; however, we can discuss these issues here and add to a revised/final version.  In this regard, one limitation is that we have thus far not integrated HALO with large-scale sampling, which would allow us to apply it to huge graphs.  A potential negative societal impact is that as the node classication accuracy with heterogeneous graphs significantly improves with models like HALO, it could be maliciously used to improve the quality of the recommendation of socially damaging products on the Internet (e.g., dangerous weapons or harmful videos on streaming websites).

---

> > ### Comment · Reviewer_Sdud · 2022-08-08
> > **Reply to authors' response**
> >
> > Thanks for the authors' response. In existing heterogeneous graph-related works [1, 2], they measure the performance of existing homogenous graph-based GNNs (e.g., GCN, GAT) on heterogeneous graphs by ignoring node types and edge types and viewing them as homogeneous graphs. From that point of view, I suggested that it would be good to compare with models solving the over-smoothing issue for homogeneous graphs. I'll keep the score.
> >
> > [1] Yun, Seongjun, et al. "Graph transformer networks." Advances in neural information processing systems 32 (2019).
> > [2] Lv, Qingsong, et al. "Are we really making much progress? Revisiting, benchmarking and refining heterogeneous graph neural networks." Proceedings of the 27th ACM SIGKDD Conference on Knowledge Discovery & Data Mining. 2021.

---

> > > ### Author Response · Authors · 2022-08-08
> > > **Reply to the response**
> > >
> > > Regarding tests with homogeneous baselines, we now understand the reviewer's intended meaning.  And for reference, the exact test the reviewer suggests has actually already been conducted in prior work [15] cited in our submission.  Specifically, from Table 4 in [15] there are head-to-head results comparing GCN and GAT with the Simple-HGN heterogeneous model.  Note that the Micro-F1 scores reported in Table 1 of our submission are obtained under the exact same settings as those from Table 4 in [15], and hence, we can readily infer that HALO achieves significantly higher performance than GCN and GAT homogeneous baselines (as well as Simple-HGN as stated in our paper).  We can clarify this in the revision for reference purposes; good suggestion.

---

### Official Review · Reviewer_zy2F · 2022-07-25

**Rating:** 7
**Confidence:** 4
**Soundness:** 3 good
**Presentation:** 4 excellent
**Contribution:** 3 good

**Summary:**

This paper proposes a systematical way of constructing energy functions for heterogenous graphs with multiple edge types and node types. This approach is justified from both the theoretical perspective and the experiments.


**Questions:**

See above.

**Limitations:**

See above.

**Strengths And Weaknesses:**

### Strengths:
* The presentation is very clear. The authors analyzed the drawbacks of direct extention of graph regularizer from single-edge type graph and proposed a term that considers the inter-edge-type correlation as a regularizer with rigorous theory analysis.
* The results of combining proximal gradient and nice trick of gradient step rescaling (L209) makes the overall procedure sound and valid.
* The paper achieves state-of-the-art results in multiple datasets.

### Weakness:
* I only have a small presentation-level comment. It would be better if the authors can move the comparing part of their proposed $y_{si}H_t-y_{s'j}$ in Eq (8) to the introduction/abstract section to let the readers know the intuition that cross-edge-type correlation is important to heterogenous graph learning. Also it can be better if the authors can visualize what the learned $M_t$ is to justify that it's not a diagonal matrix.

#### Minor:
3. L139 (ii) "the loss can be equivalently expressed with symmetric weights" is unclear to me.
4. L143 (iv) the expression that "symmetrical w.r.t. order of nodes" is confusing. $M_t$ can be asymmetrical. Can the authors explain more on this?

---

> ### Author Response · Authors · 2022-08-02
> **Response to Reviewer zy2F**
>
> Thanks for the constructive feedback and for pointing out the clarity and soundness of our proposed approach.  We address each question/comment below.
>
> ### Weakness: Small presentation-level comment
>
> We can revise the introduction to more clearly introduce upfront the importance of cross-edge-type correlations to heterogeneous graph models as the reviewer suggests. Thanks for the suggestion.
>
>
> ### Other minor questions/issues:
>
> - Regarding the comment on Line 139: Note that  $M_t = \frac{1}{2}[(M_t+M_t^T) + (M_t-M_t^T)]$ for any weight matrix $M_t$, where the first term is necessarily symmetric by design.  We then observe on Line 142 that the loss penalty involving $M_t$ decomposes into terms of the from $z^T M_t z$, where $z = (y_i - y_j)$.  However, since $z^T (M_t-M_t^T) z = 0$ for any $z$ and $M_t$, the penalty actually only depends on the symmetric part $(M_t+M_t^T)$, and hence the overall loss from Eq. (7) can be equivalently expressed with all symmetric weights, i.e., an asymmetric $M_t$ injects no additional flexibility into the model beyond its symmetric counterpart.
> - Regarding the comment on Line 143: The intended meaning is that in Eq. (7), the penalty from Line 142 is invariant to the order of the nodes, meaning $(y_i - y_j)^T M_t (y_i - y_j) = (y_j - y_i)^T M_t (y_j - y_i)$.  Hope this helps to clarify the issue.

---

> > ### Comment · Reviewer_zy2F · 2022-08-08
> > **Reply to authors' response**
> >
> > I would like to thank the authors for the detailed clarification. I would gladly keep my score and recommend acceptance for this submission.

---

### Meta-Review · Area_Chair_Qa3g · 2022-08-27

**Recommendation:** Accept
**Confidence:** Certain

**Metareview:**

This paper studies heterogeneous GNN and proposes a novel HGNN architecture HALO. It tries to make connected nodes share similar labels, with a relation-dependent compatibility matrices to resolve the mistmatch between different node types. The key is to derive novel GNN layers from the minimization steps of a relation-aware energy function.

Authors have addressed concerns on the designed choices, experiments with larger and more datasets during the rebuttal. While the idea is extended from previous works [32, 35], reviewers generally appreciate its contributions made to heterogeneous GNNs.

**Award:**

No

---

### Decision · Program_Chairs · 2022-09-14

Accept